# Cross-species cortical alignment identifies different types of anatomical reorganization in the primate temporal lobe

Nicole Eichert[1]*, Emma C Robinson[2], Katherine L Bryant[3], Saad Jbabdi[1], Mark Jenkinson[1], Longchuan Li[4], Kristine Krug[5,6,7], Kate E Watkins[8], Rogier B Mars[1,3]

[1]Wellcome Centre for Integrative Neuroimaging, Centre for Functional MRI of the Brain (FMRIB), Nuffield Department of Clinical Neurosciences, John Radcliffe Hospital, University of Oxford, Oxford, United Kingdom; [2]Biomedical Engineering Department, King's College London, London, United Kingdom; [3]Donders Institute for Brain, Cognition and Behaviour, Radboud University Nijmegen, Nijmegen, Netherlands; [4]Marcus Autism Center, Children's Healthcare of Atlanta, Emory University, Atlanta, United States; [5]Department of Physiology, Anatomy and Genetics, University of Oxford, Oxford, United Kingdom; [6]Institute of Biology, Otto-von-Guericke-Universität Magdeburg, Magdeburg, Germany; [7]Leibniz-Insitute for Neurobiology, Magdeburg, Germany; [8]Wellcome Centre for Integrative Neuroimaging, Department of Experimental Psychology, University of Oxford, Oxford, United Kingdom

**Abstract** Evolutionary adaptations of temporo-parietal cortex are considered to be a critical specialization of the human brain. Cortical adaptations, however, can affect different aspects of brain architecture, including local expansion of the cortical sheet or changes in connectivity between cortical areas. We distinguish different types of changes in brain architecture using a computational neuroanatomy approach. We investigate the extent to which between-species alignment, based on cortical myelin, can predict changes in connectivity patterns across macaque, chimpanzee, and human. We show that expansion and relocation of brain areas can predict terminations of several white matter tracts in temporo-parietal cortex, including the middle and superior longitudinal fasciculus, but not the arcuate fasciculus. This demonstrates that the arcuate fasciculus underwent additional evolutionary modifications affecting the temporal lobe connectivity pattern. This approach can flexibly be extended to include other features of cortical organization and other species, allowing direct tests of comparative hypotheses of brain organization.

*For correspondence:
nicole.eichert@psy.ox.ac.uk

## Introduction

The temporal lobe is a morphological adaptation of the brain that is unique to primates (*Bryant and Preuss, 2018*). Its origins likely include expansion of higher-order visual areas to accompany the primate reliance on vision (*Allman, 1982*). Temporal association cortex contains areas devoted to higher-level visual processing and social information processing (*Rushworth et al., 2013*; *Sallet et al., 2011*) that, in turn, rely strongly on visual information in primates (*Perrett et al., 1992*). The expanded temporal cortex in apes and humans contains several multimodal areas and areas associated with semantics and language (*Dronkers et al., 2004*; *Hickok and Poeppel, 2007*;

**eLife digest** How did language evolve? Since the human lineage diverged from that of the other great apes millions of years ago, changes in the brain have given rise to behaviors that are unique to humans, such as language. Some of these changes involved alterations in the size and relative positions of brain areas, while others required changes in the connections between those regions. But did these changes occur independently, or can the changes observed in one actually explain the changes we see in the other?

One way to answer this question is to use neuroimaging to compare the brains of related species, using different techniques to examine different aspects of brain structure. Imaging a fatty substance called myelin, for example, can produce maps showing the size and position of brain areas. Measuring how easily water molecules diffuse through brain tissue, by contrast, provides information about connections between areas.

Eichert et al. performed both types of imaging in macaques and healthy human volunteers, and compared the results to existing data from chimpanzees. Computer simulations were used to manipulate the myelin-based images so that equivalent brain areas in each species occupied the same positions. In most cases, the distortions – or 'warping' – needed to superimpose brain regions on top of one another also predicted the differences between species in the connections between those regions. This suggests that movement of brain regions over the course of evolution explain the differences previously observed in brain connectivity.

But there was one notable exception, namely a bundle of fibers with a key role in language called the arcuate fasciculus. This structure follows a slightly different route through the brain in humans compared to chimpanzees and macaques. Eichert et al. show that this difference cannot be explained solely by changes in the positions of brain regions. Instead, the arcuate fasciculus underwent additional changes in its course, which may have contributed to the evolution of language.

The framework developed by Eichert et al. can be used to study evolution in many different species. Interspecies comparisons can provide clues to how brain structure and activity relate to each other and to behavior, and this knowledge could ultimately help to understand and treat brain disorders.

*Price, 2000*). As such, understanding the evolution of temporal cortex across the primate order is a vital step to understanding primate behavioral adaptations.

Two lines of evidence are often brought to bear on differences in temporal lobe organization across humans and other primates. The first line emphasizes selective local expansions of temporal cortex and subsequent relocation of areas. Morphologically, great apes possess an extra sulcus in the temporal cortex, suggesting at the very least expansion of this part of cortex. *Mars et al. (2013)* reported a region in the middle part of the superior temporal sulcus of the macaque that shares anatomical features of the human temporo-parietal junction area located at the caudal end of the temporal cortex, suggesting a major relocation of this area. In a similar vein, *Patel et al. (2019)* suggest that expansion of the temporo-parietal junction and superior temporal sulcus gave rise to a modified ventral visual processing stream to support increased social abilities in humans. The second line of evidence emphasizes changes in the connectivity of the temporal lobe. *Rilling et al. (2008)* first suggested dramatic expansion of the arcuate fasciculus temporal cortex projections in the human, but more recent studies also emphasize increased projections of the middle longitudinal and inferior fronto-occipital fasciculi and their role in language-related processes in the human (*Catani and Bambini, 2014*; *Makris et al., 2013*; *Makris and Pandya, 2009*; *Saur et al., 2008*).

These different schools place different emphasis on what happened to temporal cortex across different primate lineages. Their results, however, should be interpreted in relation to one another as species differences in brain organization can come in many forms that can interact in unpredictable ways (*Krubitzer and Kaas, 2005*; *Mars et al., 2018a*; *Mars et al., 2017*). Dissociating such different types of species differences is challenging (*Figure 1A,B*). For instance, given an ancestral or reference state, local expansions of the cortical sheet can lead to the relocation of homologous areas between two species. As a case in point, human MT+ complex is located much more ventrally in

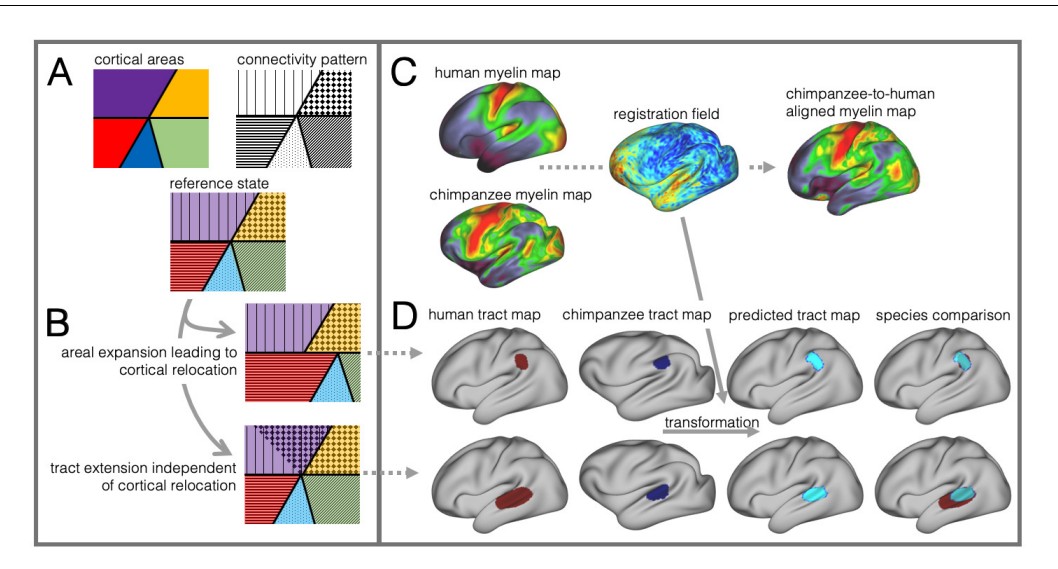

**Figure 1.** Cortical specializations. (**A**) Cortical brain organization can be described using different modalities such as brain areas defined by myelin content (top left) or the pattern of brain connections (top right). (**B**) Anatomical changes can affect both modalities differentially. Top and bottom panels show different evolutionary scenarios. (**C**) Alignment of homologous brain areas derived from myelin maps can model cortical expansion across species, here shown for human and chimpanzee. (**D**) Applying the cross-species registration field to surface tract maps allows us to distinguish evolutionary scenarios. Here shown are toy example maps of one tract that extended due to areal expansion alone (top panel) and one tract that additionally extended into new brain areas (bottom panel). Red: human tract map; dark blue: chimpanzee tract map; light blue: transformed chimpanzee tract map.

posterior temporal cortex than its macaque homolog (*Huk et al., 2002*). Such cortical relocations also affect the location of connections of these areas, but this situation is distinct from the scenario in which a tract extends into new cortical territory.

These two scenarios are illustrated in *Figure 1B*. The red area in the top panel has expanded, leading to a relocation of the blue area with respect to the purple and yellow area. The connections of the areas do not change in this scenario, resulting in a relocation of the connections of the blue area. In the bottom panel, a yellow area's tract terminations have invaded the neighboring purple territory, but this change is independent from cortical expansion. Thus, in both cases connections are located in a different place from those in the reference state, but the causes are different.

In this study we investigate to which extent species differences in temporal lobe organization are due to cortical relocation and tract extension. To be able to do this, we propose a framework to test among different forms of cortical reorganization by registering brains together into a single shared coordinate system (*Figure 1C,D*). Such an approach allows us to place different brains into a common space based on one feature and then compare the results to registration based on another feature. This 'common space' concept proved feasible in a previous study testing whether the extension of the human arcuate fasciculus (AF) compared with the macaque AF could be accounted for by differential cortical expansion between the two brains (*Eichert et al., 2019*). In the present study, we generalize this approach to develop a cross-species registration based on a multimodal surface matching algorithm (MSM, *Robinson et al., 2018*; *Robinson et al., 2013*; *Robinson et al., 2014*) to derive a cortical registration between different species.

We based our registration framework on whole brain neuroimaging data of macaque, chimpanzee and human brains. Neuroimaging allows one to acquire high-resolution data from the same brains using different modalities within a short time. The digital nature of the data allows easy manipulation, making it ideal for the present purposes (*Le et al., 1985*; *Grannell and Mansfield, 1975*; *Lauterbur, 1973*; *Le Bihan et al., 1986*; *Thiebaut de Schotten et al., 2019*). As primary modality we use surface maps derived from the cortical ribbon of T1- and T2-weighted scans, which have been shown to correlate well with cortical myelinization and which are available for all three species (*Glasser et al., 2014*; *Glasser and Van Essen, 2011*). Such 'myelin maps' can be used to

identify homologous areas across brains and species, such as primary sensory and motor cortex, which is high in myelin, and association cortex, which is low in myelin (*Glasser et al., 2014*; *Large et al., 2016*). As second modality, we use diffusion MRI tractography to reconstruct long range white matter fibers of the temporal and parietal lobes to establish its connections (*Bryant et al., 2019*; *Mars et al., 2018c*).

Given data from these two modalities, we developed an approach to reveal different types of cortical reorganization. We argue this approach is particularly suitable to study the temporal lobe, as it has well described myelin markers and cortical connections (*Glasser et al., 2014*; *Large et al., 2016*; *Mars et al., 2013*; *Ruschel et al., 2014*). First, we register the cortical surfaces of the different species to one another based on myelin maps (*Figure 1C*). This cortical alignment uses the distinction of primary and higher order areas in myelin maps as anchor points across all three species. Next, we apply this registration to overlay homologous parts of the cortex and to calculate the underlying distortions of the cortical sheet. This registration field effectively models areal expansion or contraction underlying cortical relocation of homologous areas. We then apply this registration to the cortical projection maps of temporal and inferior parietal lobe white matter tracts to assess how well the myelin-based registration can predict changes in tract projection patterns across species (*Figure 1D*). A good prediction, i.e. a high overlap of the tract maps, indicates that cortical expansion and relocation of targets zones alone can predict tact projections (*Figure 1D*, top), a poor prediction indicates that the tract is reaching new cortical territory (*Figure 1D*, bottom).

Here, we distinguish different scenarios of cortical evolution for a set of temporal and parietal white matter tracts. By applying a cross-species registration we can infer if only cortical relocation was affecting a tract's connectivity profile or if a tract is reaching into new cortical territory. A deeper understanding of species differences in brain reorganization is essential for our understanding how evolutionary specializations of the temporal lobe underlie uniquely human cognitive functions.

## Results

We set out to investigate different types of cortical reorganization affecting the temporal lobe across macaque, chimpanzee, and human brains. First, we investigated cortical relocation of brain areas by registering myelin maps of the cortical surface to one another and derived the local distortions required. Second, we used the resulting mesh deformations to transform maps of cortical projections of major white matter tracts that terminate in temporal and inferior parietal lobe. This allowed us to assess how well the myelin registration predicts actual projection maps across species or whether it cannot capture them, providing an index of tract extensions in the human brain.

### Myelin registration

We developed a surface registration between species based on myelin maps using multimodal surface matching (MSM, *Robinson et al., 2018*). *Figure 2* shows the final results of chimpanzee-to-human, macaque-to-chimpanzee, and macaque-to-human brain registrations. The cross-species registration aligns the myelin maps well, with the predicted human maps showing most of the distinctive features of the actual human myelin map (*Figure 2*, top row). Posterior areas such as V1 are well aligned, with the highest myelin evident on the medial part of the occipital cortex, having relocated quite substantially from a more lateral orientation in the macaque. The prominent myelin hot spot in the location of the MT+ complex is also noticeable. Areas where the myelin maps showed fewer distinctive features to guide the registration, such as in the prefrontal cortex, showed some differentiation between the predicted and actual human maps. Spatial correlation maps of the human myelin maps and the predicted myelin maps as well as the deformation fields underlying the registrations are provided in *Appendix 2—figure 1*.

### Tract maps

We constructed the cortical projection maps of the following tracts in all three species: Middle longitudinal fasciculus (MDLF), inferior longitudinal fasciculus (ILF), the third branch of the superior longitudinal fasciculus (SLF3), the inferior fronto-occipital fasciculus (IFO), and the arcuate fasciculus (AF) (*Figure 3A,C,E*). The human and macaque tract maps resemble those obtained in previous studies (*Mars et al., 2018c*; *Schmahmann and Pandya, 2009*) and the chimpanzee SLF3 and AF are similar

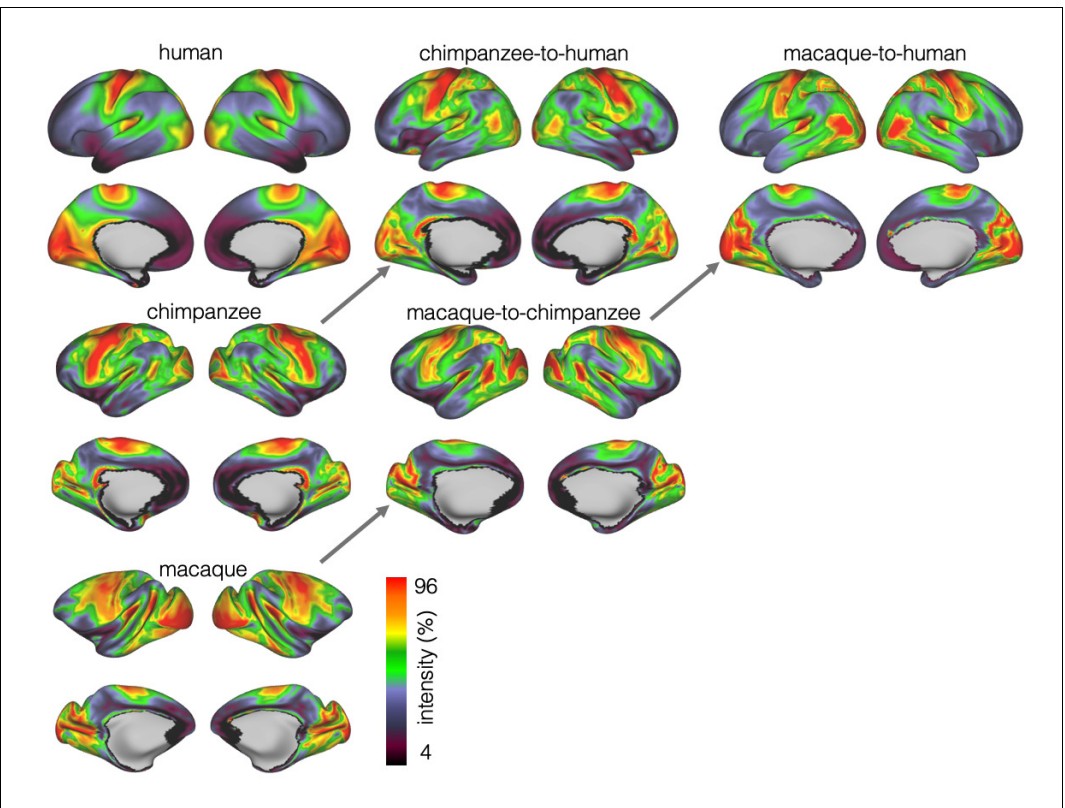

**Figure 2.** Myelin registration. Species average myelin maps (left panel) and myelin maps resampled across species after applying the MSM-derived registration.

to previous reports (*Hecht et al., 2015*; *Rilling et al., 2008*). The other chimpanzee tracts are reported here for the first time, apart from a previous exploratory study (*Mars et al., 2019*).

We applied the myelin-based surface registration to assess whether the cortical relocation demonstrated in the myelin registration above fully explains the changes in tracts. *Figure 3* shows actual

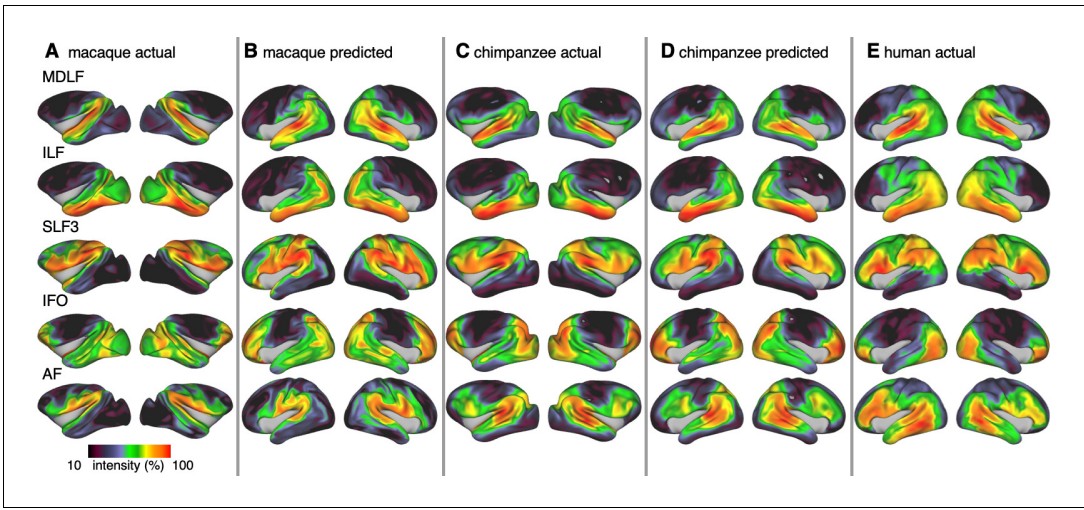

**Figure 3.** Actual and predicted tract maps. The intensity in the tract maps reflects the probability of a tract's termination on the cortical surface as derived from tractography. Actual tract maps of macaque (**A**), chimpanzee (**C**) and human (**E**). **B** and **D** show the tract maps in human space, predicted by the myelin-based registration for macaque and chimpanzee.

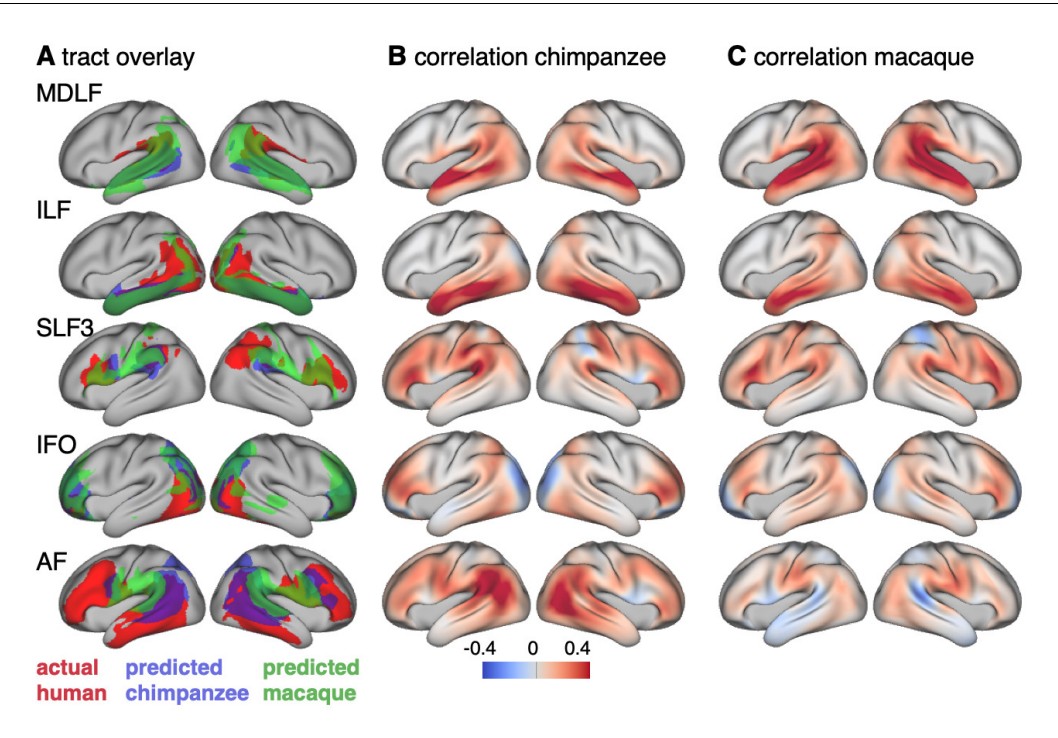

**Figure 4.** Cross-species comparison results. (**A**) Thresholded actual human tract maps (red) and tract maps predicted by the myelin-based registration for chimpanzee (blue) and macaque (green) (thresholds (t): MDLF: t = 0.7, ILF: t = 0.7, SLF3: t = 0.85, IFO: t = 0.75, AF: t = 0.75). (**B**), (**C**) Weighted correlation maps of actual human map and predicted chimpanzee and macaque map.

and predicted tract maps. For visual assessment, a thresholded overlay of actual human and predicted tract maps is shown in *Figure 4A*. As described above, we focus on a description of temporo-parietal cortex given the multiple competing theories of its reorganization in different primate lineages.

We assessed the success of the myelin registration in predicting the tract projections in a number of ways. First, weighted correlation maps provide a visualization of the local quality of the prediction (*Figure 4B,C*). A high value means that the myelin registration alone is sufficient to predict a tract's projection in this part of the brain. A low correlation value indicates that reorganization of a tract's connectivity pattern took place in addition to cortical relocation modelled by the myelin registration. Second, the Dice coefficient of similarity provides a more general measure of similarity between the predicted and actual tract maps, where a Dice coefficient of '1' indicates perfect overlap and thus no tract extension into areas other than would be predicted by cortical relocation assessed using the myelin map registration. Finally, we calculated a 'tract extension ratio' that indicates how much of the actual human tract projections extends into parts of the surface not predicted, where a value of >1 indicates a tract extension into novel territory. Both the Dice coefficients and tract extension ratios were computed for thresholded tract maps defined by the human tract map covering 40% of the brain's surface, but the resulting pattern of values is robust across a range of thresholds (see *Appendix 3—figure 2*).

In general, it can be observed that the myelin-based registration can predict the tract maps well in both hemispheres, with the notable exception of AF and to a lesser extent ILF and SLF3 (*Figure 4*). AF in particular shows the lowest Dice coefficient and the highest extension ratio (*Figure 5A,B*), indicating that this tract's differential projections in the human brain are not merely due to relocation of areas. The maps for macaque and chimpanzee are overall predicted to a similar degree, as can be seen in the overlay and correlation maps, with the exception of AF (*Figure 4*). The effect for AF is captured in the Dice coefficients and tract extension measures (*Figure 5A,B*).

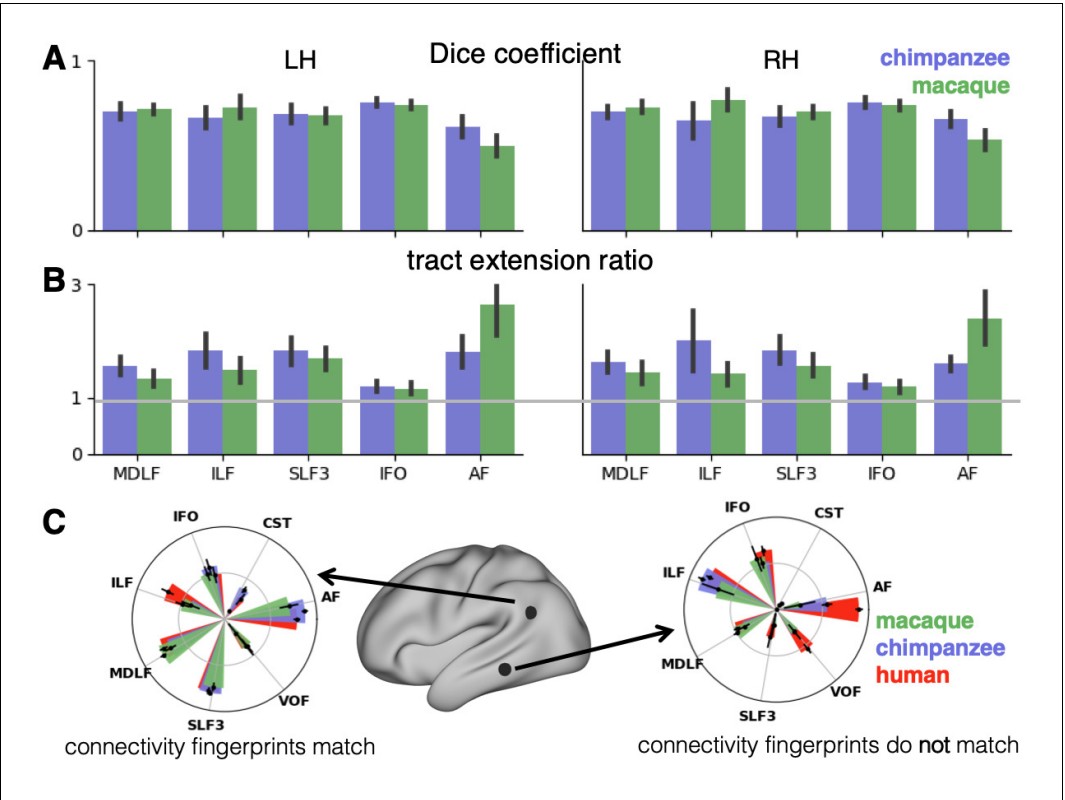

**Figure 5.** Quantification of cross-species comparisons. Dice coefficients of overlap (**A**) and tract extension ratios (**B**) computed from actual thresholded human tract maps and tract maps predicted by the other species. Shown are mean and standard deviation derived from all pairs of human (n = 20) and macaque or chimpanzee (n = 5) subjects in left (LH) and right (RH) hemisphere. (**C**) Connectivity fingerprints at two vertices in inferior parietal and in temporal lobe derived from the intensity values of an extended set of tract maps. Shown are mean and standard deviation (human: n = 20, chimpanzee and macaque: n = 5).

The online version of this article includes the following source data for figure 5:

**Source data 1.** Numerical data underlying the connectivity fingerprint shown in *Figure 5C*.

A two-way statistical analysis was performed in both hemispheres to assess the effect of species and tract on the extension ratios. In the left hemisphere, there was no significant main effect of species ($F_{(1, 179)}=1.38$, p=0.47), but a highly significant main effect of tract ($F_{(4, 792)}=565.00$, p<0.001) and a highly significant interaction effect of species and tract ($F_{(4, 792)}=207.73$, p<0.001). In the right hemisphere, we found a significant main effect of species ($F_{(1, 179)}=16.76$, p<0.001) as well as a highly significant effect of tract ($F_{(4, 792)}=261.94$, p<0.001) and a highly significant interaction effect ($F_{(4, 792)}=225.70$, p<0.001). We will discuss the various tracts in more detail below.

The myelin-based registration results in good prediction for tract projections in the temporo-parietal cortex. The actual human tract terminations of MDLF span the superior temporal gyrus and reach the inferior parietal cortex (*Figure 4A*). In the macaque and chimpanzee, the actual MDLF terminates in superior temporal gyrus but reaches only to a small part of the inferior parietal cortex. When applying the myelin-based registration, macaque and chimpanzee MDLF are both predicted to reach a comparable portion of the human temporal lobe and parts of both angular and supramarginal gyri of the inferior parietal lobe (*Makris et al., 2013*). This overlap is captured in the weighted correlation maps, which have high values in the temporal lobe (*Figure 4B,C*). The Dice coefficients for the chimpanzee and macaque MDLF are high and the extension ratio is close to one indicating no tract extension in addition to cortical expansion (*Figure 5A,B*).

A similar observation can be made for the posterior terminations of SLF3 and IFO. The myelin-based registration can predict the parietal cortical projections to a large degree. The predicted cortical terminations of the tract show a strong overlap with the actual human tract terminations. In line

with the overlay maps and the weighted correlation maps, the Dice coefficients are relatively high. Taken together, this suggests that expansion and relocation of brain areas are largely sufficient to model the posterior cortical terminations of SLF3 and IFO, while extension of the tract's connectivity pattern plays a minor role in explaining the species differences.

Predicted ILF terminations show that the expected occipito-temporal connection can be modelled well (*Catani and Thiebaut de Schotten, 2008*). The extension ratio for ILF is elevated indicating that there is some remaining tract extension that has not been modelled by cortical relocation. This is also reflected in the tract overlay (*Figure 4A*), which shows that human ILF has more extended posterior projections than predicted by the myelin registration. Thus, although the overall architecture of the tracts is well predicted, this tract seems to have extended into new cortical territory in the human lineage.

The clearest case of a tract extension in the human brain was presented by AF. Human AF reaches anteriorly to inferior and dorsal frontal areas. The posterior projections of human AF reach into middle and inferior temporal cortex. The chimpanzee posterior terminations are in inferior parietal lobe and superior temporal cortex and in the macaque, the temporal projections reach superior temporal areas. For AF, the myelin-based registration does not provide a good prediction of the tract map across species, especially for the macaque. The correlation map for the chimpanzee shows low correlation along the temporal lobe and for the macaque, correlation values in temporal lobe are extremely low. AF has the lowest Dice coefficient and the extension ratio is high, especially in the macaque, which is in line with overlay and correlation maps. The 'failure' of the myelin registration in the temporal lobe indicates that extension and relocation of cortical areas is not sufficient to explain the posterior tract projections of AF, but that the tract extended into new cortical territory in the temporal lobe.

## Connectivity fingerprints

To further characterize the effects in the predicted tract maps, we obtained connectivity fingerprints at two representative vertices on the left brain surface: One in inferior parietal lobe, where most tracts are predicted well and one in the middle temporal gyrus, where we observed strong species differences, in particular for AF. The connectivity fingerprints were derived based on the intensity values in the actual and predicted tract maps using an extended set of seven tracts to give a more detailed picture (see *Appendix 3—figure 1*). *Figure 5C* demonstrates that in the inferior parietal lobe, the connectivity fingerprint of actual human and predicted chimpanzee and macaque tract maps are highly similar, except for a small increase in the intensity of ILF indicating tract extension as discussed above. This indicates that the myelin-derived registration can predict this area's connectivity profile well, despite the local expansion of the cortical sheet. For the temporal vertex, however, there is a strong mismatch regarding the connectivity profile, in particular regarding the intensities for AF. This indicates that the connectional fingerprint of this temporal area is different in the human than would be predicted purely based on cortical relocation. Thus, the connectivity fingerprints of the two representative areas match the pattern of species differences that emerged from the results above.

## Discussion

The goal of this study was to study brain specializations of the temporal lobe across multiple primate species in the context of two forms of cortical reorganization: cortical relocation due to local expansions and extensions of tracts into new cortical territory (*Figure 1*). For this reason, we developed a cross-species surface registration method based on cortical myelin content, which gives us an index of how cortical areas have relocated during evolution. In a subsequent step, we tested if cortical relocation can predict the connectivity patterns of a set of tracts across species. We showed that cortical expansion and resulting relocation of brain areas alone provide a good prediction of several tracts' terminations in posterior temporal and parietal cortex. In the case of AF in particular, we showed that an additional change in brain architecture was extension of the tract into new cortical areas independent of cortical expansion and resulting relocation.

As pointed out in the introduction, different lines of evidence on temporal reorganization across primate species have emphasized either the expansion and subsequent relocation of brain areas or changes in temporal lobe connectivity. Both approaches are valuable and we here demonstrate that

both are applicable to different parts of the temporal lobe. Previous work has suggested expansion and relocation of areas in posterior temporal cortex and adjacent parietal cortex (*Mars et al., 2013*; *Patel et al., 2019*), which is consistent with the ventral location of MT+ complex in humans compared to other primates. Similar reorganizations have been suggested by *Haak et al. (2018)*, who proposed modifications of visual-temporal pathways in the human. These major relocations were captured by our myelin registration and demonstrated to predict certain features of white matter tracts, such as the posterior projections of MDLF.

The aim of our research is not to find the 'best' species registration, but to shed light on brain evolution by studying where registrations based on different modalities disagree. Many aspects of brain architecture can be modified during the course evolution and each individual aspect would provide us with a unique registration. The present study represents only one example of how cortical specializations can be studied by comparing cross-species registrations of different modalities. Testing the effect of a myelin registration on connectivity is not meant to imply pre-eminence of myelin over connectivity. In fact, the reversed approach, i.e. to derive a registration based on individual tract maps and then test their differential effect on transformed myelin maps, would be highly informative. We chose myelin here as the primary feature, because we wanted to test the specific hypothesis that some tracts expanded while others simply followed relocation of areas across the cortex. Thus, we used a measure that could index the relocation (myelin) and test it on our measure of interest (the tracts).

Separate from cortical relocation, changes in connectivity between humans and other species have been described for a variety of association tracts, including the temporal projections of AF into middle temporal gyrus (*Eichert et al., 2019*; *Rilling et al., 2008*), the frontal projections of SLF3 (*Hecht et al., 2015*), and expansion of the ventral route consisting of MDLF and IFO (*Forkel et al., 2014*; *Makris et al., 2013*; *Makris et al., 2009*). Both of these latter tracts have been suggested to play a role in language functions in the human brain (*Catani and Bambini, 2014*; *Hagoort, 2016*; *Makris et al., 2009*; *Makris and Pandya, 2009*; *Saur et al., 2008*).

In the case of MDLF, the human tract projections can be predicted well by the cortical relocation model. This indicates that the pattern of cortical terminations changed according to the general expansion and relocation of brain areas, without additional extension into new regions of the brain. For other tracts reaching to temporo-parietal areas, such as ILF, IFO and SLF3, the posterior tract projections can also be modelled well, despite the large distortions of target areas within the cortical sheet. These tracts seem to follow the evolutionary scenario described in the upper row in *Figure 1B,D*, where a tract's extension in the human brain can be explained by relocation of areas along the cortical sheet. This does not necessarily mean that the tracts have not been recruited for new functions, but the type of change is different from that of tracts such as AF.

AF showed the lowest consistency across species when applying the myelin-based registration. Dice ratio and extension ratio reflect the increased tract termination especially into the temporal lobes, which can be observed in the tract maps. This points to further evolutionary adaptations that specifically affected the connectivity pattern of AF independent of cortical relocation, the scenario described in the bottom figure in *Figure 1B,D*. Our result is consistent with previous accounts in the literature (*Ardesch et al., 2019*; *Eichert et al., 2019*; *Rilling et al., 2008*), but the approach described here enabled us to formally test this hypothesis in the wider context of cortical reorganization across three primate species and to quantify the species differences. Importantly, extension of a tract into new cortical territories alters the unique connectivity fingerprint of the innervated areas, which profoundly changes the computational capabilities that area supports (*Mars et al., 2018b*). Being able to dissociate different modifications of brain architecture can inform us about how temporal lobe specializations link to uniquely human higher cognition (*Qi et al., 2019*; *Roelofs, 2014*; *Schomers et al., 2017*).

Apart from modifications of AF, we also noted some minor extensions of ILF into temporo-parietal cortex. ILF's extension in the human brain is consistent with reports that that showed a split of this tract into multiple subtracts due to the expansion of parts of the temporal cortex, including the fusiform gyrus (*Latini et al., 2017*; *Roumazeilles et al., 2019*). This extensions could be related to the increase of cortical territory related to processing social information, such as social networks and faces (*Noonan et al., 2018*; *Sallet et al., 2011*). Previously, it has been shown that parietal SLF3 projections are most prominent in the human brain, which has been linked to our unique capacity of social learning (*Hecht et al., 2013*). We show that species differences in the posterior projections of

SLF3 can be mostly explained by local expansion of the posterior temporal and parietal cortex. Similarly, we show that cortical expansion can model the terminations of MDLF, a tract, which has been linked to visuospatial and integrative audiovisual function (*Makris et al., 2013*). Our results thus suggest that SLF3 and MDLF didn't undergo additional evolutionary modifications that affected their posterior terminations.

The MSM framework we adopted is ideally suited to work with multimodal descriptors of the cerebral cortex. It has become a vital tool for human surface registration (*Abdollahi et al., 2014*; *Garcia et al., 2018*; *Glasser et al., 2016*) and here we demonstrated its utility for cross-species research. With the presented surface matching method, we showed that a registration based on T1w/T2w MRI data can match critical landmarks across species. We have referred to these maps as 'myelin maps' in accordance with other studies in the literature (*Glasser et al., 2014*; *Large et al., 2016*) but it should be noted that this is a heuristic. T1w/T2w maps are sensitive to other features than myelin and other sequences are sensitive to aspects of cortical myelin (*Lutti et al., 2014*). The crucial point is that the maps we employed here are similar across species, allowing us to compare like with like (*Glasser et al., 2014*). Projecting data of different modalities to a surface representation is a useful tool for comparative neuroscience. It allows us to visualize the different modalities within the same cortical sheet and to compare topologies on this 2D surface, which opens a wide array of mathematical tools. A similar approach has been taken to investigate the relationship between gene expression and myelin content of the cortex (*Burt et al., 2018*) and gradients of change across multiple modalities of brain organization (*Blazquez Freches et al., 2020*; *Huntenburg et al., 2018*).

The presented approach can be flexibly modified to include a variety of cortical features, which can be compared across species. Myelin does not provide high contrast in the large human frontal cortex and, as such, it is difficult to provide a good registration in frontal areas. Furthermore, the effects we report can only be reliably interpreted within the spatial resolution of brain areas. More fine scale species differences and homology assignments are not possible with the data shown here. However, the current method can be generalized to any modality of cortical organization, so future studies can incorporate modalities that have greater contrast in this part of the brain such as neurite orientation dispersion and density imaging (NODDI) measures (*Zhang et al., 2012*) and resting state fMRI networks (*Vincent et al., 2007*).

In sum, here we present a framework for analyzing structural reorganization of the temporo-parietal cortex across different primate brains. We dissociated cortical relocation of areas due to local expansion and modifications of white matter tract connectivity. Future work will expand this approach not only to different modalities, but also to a much wider range of species, which is now becoming increasingly possible due to the availability of multi-species datasets (*Heuer et al., 2019*; *Milham et al., 2018*). This provides a crucial step towards the understanding of phylogenetic diversity across the primate brain.

## Materials and methods

### Key resources table

| Reagent type (species) or resource | Designation | Source or reference | Identifiers | Additional information |
|---|---|---|---|---|
| Software, algorithm | FSL | http://fsl.fmrib.ox.ac.uk/fsl/ | RRID:SCR_002823 | |
| Software, algorithm | FreeSurfer | http://surfer.nmr.mgh.harvard.edu/ | RRID:SCR_001847 | |
| Software, algorithm | MSM | https://fsl.fmrib.ox.ac.uk/fsl/fslwiki/MSM | RRID:SCR_002823 | MSM is available as part of FSL. Code for MSM using higher-order smoothness constrains is available online at https://www.doc.ic.ac.uk/~ecr05/MSM_HOCR_v2/ |
| Software, algorithm | Connectome Workbench | http://www.nitrc.org/projects/workbench | RRID:SCR_008750 | |

## Human data and pre-processing

Human data were acquired in 20 subjects (12 females, 18–40 years) on a 3T Siemens Prisma scanner with a 32-channel head coil. The study was approved by the Central University (of Oxford) Research Ethics Committee (CUREC, R55787/RE001) in accordance with the regulatory standards of the Code of Ethics of the World Medical Association (Declaration of Helsinki). All participants gave informed consent to their participation and were monetarily compensated for their participation.

High-resolution structural images were acquired using a (MPRAGE) T1w sequence ( TR = 1900 ms; TE = 3.97 ms; flip angle = 8°; 192 mm FoV; voxel size 1 mm isotropic) and (SPC) T2w sequence (TR = 3200 ms; TE = 451 ms; 256 mm FoV; voxel size 1 mm isotropic; Grappa factor = 2). Diffusion-weighted (DW) MRI data were acquired in the same subjects using a sequence from the UK Biobank Project (*Miller et al., 2016*). In brief, we used a monopolar Stejskal-Tanner diffusion encoding scheme (*Stejskal and Tanner, 1965*). Sampling in *q*-space included two shells at *b* = 1000 and 2000 s/mm$^2$ (voxel size 2 mm, MB = 3). For each shell, 50 distinct diffusion-encoding directions were acquired (covering 100 distinct directions over the two *b*-values). Five *b* = 0 images were obtained together with additional three *b* = 0 images with the phase-encoding direction reversed.

T1w and T2w scans were pre-processed using the HCP-pipeline (*Glasser et al., 2013*) cloned from the 'OxfordStructural' - fork (https://github.com/lennartverhagen/Pipelines). The processing pipeline includes automatic anatomical surface reconstruction using FreeSurfer and provides measures of sulcal depth and surface maps of cortical myelin content (*Fischl, 2012*; *Jenkinson et al., 2012*). The mean image of the T1w scans was divided by the mean image of the T2w scans to create a T1w/T2w image. The bias corrected T1w/T2w-ratio was mapped onto the mid-thickness surface using Connectome Workbench command-line tools. We refer to this surface map as T1w/T2w 'myelin map' (*Glasser et al., 2014*; *Glasser and Van Essen, 2011*). In order to create a human average myelin map, the subject's individual myelin maps were aligned using MSM. The myelin alignment was initialized using alignment based on maps of sulcal depth (a table with parameters is provided in *Supplementary file 1*). To create the species average maps, we used an implementation of MSM that optimizes based on a first-order (pairwise) cost function to penalize against distortions, given that no excessive distortions were expected. Human volume data were registered to the Montreal Neurological Institute standard space (MNI152) and surface data was transformed to a surface template space (fs_LR).

## Chimpanzee data and pre-processing

In vivo chimpanzee structural MRI and DW-MRI data were obtained from the National Chimpanzee Brain Resource (www.chimpanzeebrain.org). Data were acquired at the Yerkes National Primate Research Center (YNPRC) at Emory University through separate studies covered by animal research protocols approved by YNPRC and the Emory University Institutional Animal Care and Use Committee (approval no. YER-2001206). Both structural MRI and DWI-MRI data were collected on a Siemens 3T Trio Scanner (Siemens Medical System, Malvern, PA, USA). These chimpanzee MRI scans were obtained from a data archive of scans obtained prior to the 2015 implementation of U.S. Fish and Wildlife Service and National Institutes of Health regulations governing research with chimpanzees. All the scans reported in this publication were completed by the end of 2012.

T1w/T2w myelin maps were obtained from a group of 29 adult chimpanzees (all female), scanned at 0.8 mm isotropic resolution (*Donahue et al., 2018*; *Glasser et al., 2014*; *Glasser et al., 2012*). T1w and T2w scans were processed using a modified version of the HCP-pipeline (*Glasser et al., 2013*). DW-MRI data were obtained in a subset of five individuals. Acquisition and pre-processing was previously described (*Li et al., 2013*; *Chen et al., 2013*; *Mars et al., 2019*). Two DW images (TR = 5900 ms; TE = 86 ms; 41 slices; 1.8 mm isotropic resolution) were acquired using a single-shot spin-echo echo planar sequence for each of 60 diffusion directions (*b* = 1000 s/mm$^2$), each with one of the possible left–right phase-encoding directions and four repeats, allowing for correction of susceptibility-related distortion. For each repeat of diffusion-weighted images, five images without diffusion weighting (*b* = 0 s/mm$^2$) were also acquired with matching imaging parameters.

Chimpanzee volume and surface data were registered to a standard space template based on 29 chimpanzee scans acquired at the YNPRC (*Donahue et al., 2018*). A species average myelin map from the 29 chimpanzees was derived using MSM as described for the human.

## Macaque data and pre-processing

Ex vivo DW-MRI data were obtained from four rhesus macaques (one female, age at death: range 4–14 years) using a 7T magnet with Agilent Directive (Agilent Technologies, Santa Clara, CA, USA). Data acquisition and DW-MRI pre-processing have been previously described in detail (*Eichert et al., 2019*; *Folloni et al., 2019*). Data were acquired using a 2D diffusion-weighted spin echo multi slice protocol with single line readout (DW-SEMS; TE = 25 ms; TR = 10 s; matrix size: 128 × 128; resolution 0.6 mm; number of slices: 128; slice thickness: 0.6 mm). Nine non-diffusion-weighted ($b$ = 0 s/mm$^2$) and 131 diffusion-weighted ($b$ = 4000 s/mm$^2$) volumes were acquired with diffusion encoding directions evenly distributed over the whole sphere, except in one monkey were seven non-diffusion-weighted images and 128 diffusion directions were collected. This protocol and similar ones have previously shown to be sufficient for comparison with in vivo human data (see for example: *D'Arceuil et al., 2007*; *Dyrby et al., 2011*; *Eichert et al., 2019*; *Mars et al., 2016*).

Additionally, ex vivo data from one male macaque were obtained (*de Crespigny et al., 2005*) and pre-processed as described previously (*Jbabdi et al., 2013*). Relevant imaging parameters for DW-MRI data were: 4.7T Oxford magnet equipped with BGA 12 gradients; 3D segmented spin-echo EPI 430 μm isotropic resolution, eight shots, TE = 33 ms, TR = 350 ms, 120 isotropically distributed diffusion directions, $b$ = 8000 s/mm$^2$. Despite the different scanning parameters, data quality was appropriate to allow pooling of the ex vivo data sets. In vivo data from the same macaque subjects was not available.

To obtain macaque T1w/T2w myelin maps, in vivo T1w and T2w scans data were obtained from a previous study on five separate rhesus macaques (four females, age range 3.4 years - 11.75 years). Data acquisition and pre-processing of the macaque data have been described previously (*Bridge et al., 2019*; *Large et al., 2016*). Procedures of the in vivo macaque data acquisition were carried out in accordance with Home Office (UK) Regulations and European Union guidelines (EU directive 86/609/EEC; EU Directive 2010/63/EU).

Macaque surface reconstruction and average myelin maps were derived as described for the human. Macaque volume and surface data were registered to a standard space, which is based on data from 19 macaques acquired at YNPRC (*Donahue et al., 2018*; *Donahue et al., 2016*).

## Myelin-based surface registration

Our aim was to derive a cross-species registration to model expansion and relocation of cortical brain areas. Therefore, we performed registration based on average surface myelin maps in the three species using MSM with higher-order smoothness constraints (*Ishikawa, 2014*; *Robinson et al., 2018*). We derived a transformation of the cortical surface so that homologous myelin landmarks across species matched. The general processing steps were as follows, but a more detailed description of the methodology and an explanatory figure are provided in Appendix 1.

We obtained a 'chimpanzee-to-human' and a 'macaque-to-chimp' registration. A 'macaque-to-human' registration was derived as a concatenation of both registration stages to minimize the between-species distortions needed. As input for the registration we used the species average myelin maps and we performed the registration for both hemispheres separately.

In general, the registration was derived using two stages. The first stage was based on three regions-of-interest (ROIs) to handle the gross distortions that are involved in matching myelin landmarks across species. Two ROIs captured the highly myelinated precentral motor cortex (MC) and MT+ complex and a third ROI covered the medial wall (MW). We used MSM to obtain a registration so that the ROIs are roughly matched across species. In the second stage, the ROI-based registration was used as initialization for the subsequent alignment of the whole-hemisphere myelin maps. To derive a macaque-to-human registration, we resampled the average macaque myelin map to chimpanzee space using the MSM-derived macaque-to-chimpanzee registration. Then we aligned the resampled macaque map in chimpanzee space with that of the human and used the chimpanzee-to-human registration as initialization.

The quality of the registration performance was assessed by computing a local spatial correlation between the human myelin map and the result of the chimpanzee and macaque registration. Furthermore, we visualized the deformations underlying the registration in form of a surface distortion map. The methods and results for these two analyses are provided in Appendix 2. .

## Tractography

Human and chimpanzee DW-MRI data were pre-processed using tools from FDT (FMRIB's Diffusion Toolbox, part of FSL 5.0 [*Smith et al., 2004*]). We applied the TOPUP distortion correction tool followed by eddy-current distortion and motion correction (*Andersson et al., 2003*; *Andersson and Sotiropoulos, 2016*) as implemented in FSL. Macaque ex vivo DW-MRI data were processed using tools from FSL as implemented in an in-house MR Comparative Anatomy Toolbox (Mr Cat, www.neuroecologylab.org).

Pre-processed DW-MRI images were processed by fitting diffusion tensors (FSL's DTIFIT [*Behrens et al., 2003*]) and by fitting a model of local fiber orientations including crossing fibers (FSL's BedpostX; *Behrens et al., 2007*; *Jbabdi et al., 2012*). Up to three fiber orientations per voxel were allowed. Tractography was performed using FSL's probtrackx2. Registration warp-fields between each subject's native space and standard space were created using FSL's FNIRT (*Andersson et al., 2007*).

We performed tractography of the following tracts: Middle longitudinal fasciculus (MDLF), inferior longitudinal fasciculus (ILF), the third branch of the superior longitudinal fasciculus (SLF3), the inferior fronto-occipital fasciculus (IFO), and the arcuate fasciculus (AF). Placement of seed, waypoint, and exclusion masks was based on previous studies, in order to reconstruct known pathways for these tracts in all three species (human and macaque: *de Groot et al., 2013*; *Mars et al., 2018c*, protocols for AF: *Eichert et al. (2019)*; chimpanzee: *Bryant et al. (2018)*). Masks were drawn in standard space and warped to native subject diffusion MRI space for probabilistic tractography. The resulting tractograms were normalized by dividing each voxel's value by the total number of streamlines that successfully traced the required route ('waytotal'). To decrease computational load for further processing all tractograms were down-sampled (human: 2 mm, chimpanzee: 1.5 mm, macaque: 1 mm). In addition, tractography and surface-based analysis was performed for cortico-spinal tract (CST) and vertical occipital fasciculus (VOF). Results for all tracts are reported in *Appendix 3—figure 1*.

## Surface tract maps

To assess which part of the cortical grey matter might be reached by the tracts, we derived the surface representation of each individual tractogram using a matrix multiplication method described in *Mars et al. (2018b)*; *Appendix 1—figure 1B(2)*. We calculated whole-hemisphere vertex-wise connectivity matrices, tracking from the 20k-vertices mid-thickness surface to all voxels in the brain. These matrices were computed for both hemispheres and each subject individually in the three species. In the macaque we used the five subject's average mid-thickness in standard space as input for the computation instead of individual surfaces.

To rebalance the weights in the tracts to be more homogenous, connectivity values were weighted by the distance between vertex and voxel. A distance matrix across all vertices of the mid-thickness surface and all brain voxels was computed using MATLAB's pdist2-function resulting in a matrix of the same size as the connectivity matrix. Each element in the connectivity matrix was then divided by the corresponding value in the vertex-to-voxel distance matrix. To decrease data storage load (approximately 10 GB per matrix) the weighted connectivity matrices of the five subjects were averaged for each hemisphere and species.

To visualize a tract's surface representation, we multiplied the averaged connectivity matrix with a tract's tractogram ('fdt_paths'). We refer to the tract surface representation here as 'tract map'. The approach described above decreases gyral bias in the resulting tract map notably when compared to surface-based tractography or surface projections of the tractogram. However, the method introduced spurious effects on the medial wall and insular cortex, which are generally not well captured in the tract map. Given that both areas are not of interest in this study, they were masked out for further analysis. Tract maps were derived for each subject and both hemisphere separately. Individual surface maps were smoothed on the mid-thickness surface (human: 4 mm kernel (sigma for the gaussian kernel function), smoothing on individual surface; chimpanzee: 3 mm kernel, smoothing on average surface; macaque: 2 mm kernel, smoothing on average surface), logNorm-transformed and averaged across subjects.

## Predicted tract maps

Next, we tested if our myelin-based registration can be used to predict the tract maps across species. We resampled individual chimpanzee and macaque tract maps to human space using the macaque-to-human and the chimpanzee-to-human registration (*Appendix 1—figure 1B(3)*). Intensity values in actual and predicted tract maps ranged from 0 to 1. We averaged all predicted tract maps and displayed the average map onto a human average surface (Q1-Q6_R440). For visual inspection we also assessed and showed thresholded tract maps. Thresholds were chosen different for each tract, ranging from 0.6 to 0.85, so that the most characteristic termination is visible.

To visualize and quantify the prediction of macaque and chimpanzee tracts in human space, we derived weighted whole-hemisphere local correlation maps of the human map and the map predicted by macaque or chimpanzee. The local correlation map was computed using a sliding window around every vertex on the sphere (diameter 10 cm for all three species) using MATLAB's corrcoef-function ( Mathworks, Natick, MA). We used a search kernel of 40° that corresponds to a circular search window with a radius of approximately 7 cm. The correlation map was modified to up-weight the brain areas where the tract is represented on the surface. A weighting mask was derived by multiplication of the intensities in the actual human tract map and the other species' predicted map. The values for the weighted correlation map are thus high in parts of the brain where both actual human and predicted tract show a termination, and where the spatial patterns of intensity values correlate. Weighted correlation maps were derived for each pair of 20 human subjects and five subjects of the other species. As result figure we display the averaged correlation map onto the human average surface.

In order to quantify how well a tract is predicted, we computed Dice coefficients of similarity (*Dice, 1945*), which quantifies the amount of overlap of the tract maps. The metric was derived for each pair of 20 human subjects and five subjects of the other species. The Dice coefficient was computed for the binarized and thresholded actual human tract map and the map predicted by the other species. The threshold was chosen for each tract individually so that 40% of surface vertices were covered by the human tract map. The same threshold was applied to the macaque and chimpanzee map.

As a quantification of tract extension, we computed the ratio of the number of vertices covered by the thresholded human tract map and the number of vertices covered by both the human and the other tract map. To confirm that the pattern of values is robust, both Dice coefficients and tract extension ratios were computed for a range of percentages of surface coverage and data for a coverage of 20%, 30% and 50% is provided in *Appendix 3—figure 2*.

The differences in tract extension ratios at a surface coverage of 40% were assessed in a nonparametric permutation test implemented in PALM (*Winkler et al., 2014*) using 5000 permutations. We constructed a mixed-effects model matrix using R software (*R Development Core Team, 2015* Core *p*-values were corrected for family-wise error over multiple contrasts.

We performed additional control analyses to assess if the observed effects of tract expansion correlate with potential sources of confounds arising from our connectivity measures and the myelin-driven registration. The methodology and results of these control analyses are reported in Appendix 4.

## Connectivity fingerprints

We characterized the effect of cortical expansion on brain connectivity using the concept of connectivity fingerprints (*Passingham et al., 2002*). In brain areas where cortical expansion can explain the human connectivity pattern, actual and predicted tract maps will have similar intensity values. In brain areas where the connectivity profile was further modified due to tract extensions, the intensity values of actual and predicted tract maps will show a discrepancy. By computing the intensity values of multiple tract maps in a brain area, we can derive a characteristic profile of values that can be understood as connectivity fingerprint of this area. In brain areas, where tract extension happened in addition to cortical expansion, we expect to observe a difference between actual human connectivity profile and the predicted connectivity profile. We manually selected two representative vertices and derived their actual and predicted connectivity profile: One vertex in the inferior parietal lobe, where we expect the intensity values of actual and predicted tract maps to be similar and one in the middle temporal gyrus, where we expect to find differences in actual and predicted tract maps. The whole

set of tracts investigated (CST, MDLF, VOF, IFO, ILF, SLF3 and AF) was included to give a more detailed estimate of the connectivity fingerprint.

## Code availability statement

Availability of software used in the present study is provided in the Key Resources Table. Processing code is openly available from the Wellcome Centre for Integrative Neuroimaging's GitLab at https://git.fmrib.ox.ac.uk/neichert/project_MSM (*Eichert, 2020*; copy archived at https://github.com/elifesciences-publications/project_msm).

## Data availability overview

| Data set | Reference for original data paper | Availability |
|---|---|---|
| Human in-vivo diffusion MRI data and myelin maps | present study | Anonymised raw data is openly available for download via OpenNeuro. Accession code: ds002634 (version 1.0.1), project_larynx (https://openneuro.org/datasets/ds002634) |
| Chimpanzee in-vivo diffusion MRI data | (*Chen et al., 2013*) | Available from the National Chimpanzee Brain Resource (http://www.chimpanzeebrain.org/). Data from the following subjects were used: Bo, Cheetah, Lulu, Wenka, Foxy. |
| Chimpanzee in-vivo myelin maps | (*Glasser et al., 2014*) | Raw data available from the National Chimpanzee Brain Resource (http://www.chimpanzeebrain.org/). Data from all 29 subjects were used. |
| Macaque ex-vivo diffusion MRI data (4 macaques) | (*Folloni et al., 2019*) | Source data available from the PRIMatE Data Exchange (PRIME-DE) resource (http://fcon_1000.projects.nitrc.org/indi/indiPRIME.html4). Dataset: University of Oxford WIN Macaque PM |
| Macaque in-vivo myelin maps | (*Bridge et al., 2019*; *Large et al., 2016*) | Data of four monkeys freely available at: https://gin.g-node.org/hbridge_oxford/brainwithoutv1. Data of the fifth monkey available upon request. |

## Acknowledgements

NE is a Wellcome Trust Doctoral student in Neuroscience at the University of Oxford [ 203730/Z/16/Z]. The project was supported by the NIHR Oxford Health Biomedical Research Centre. The Wellcome Centre for Integrative Neuroimaging is supported by core funding from the Wellcome Trust [203139/Z/16/Z]. The work of RBM is supported by the Biotechnology and Biological Sciences Research Council (BBSRC) UK [BB/N019814/1] and the Netherlands Organization for Scientific Research NWO [452-13-015]. The work of ECR was supported by the Academy of Medical Sciences/the British Heart Foundation/the Government Department of Business, Energy and Industrial Strategy/the Wellcome Trust Springboard Award [SBF003\1116]. The work of KLB is supported by a Marie Sklodowska-Curie Postdoctoral Research Fellowship from the European Commission [750026]. The work of SJ is supported by the Medical Research Council UK [MR/L009013/1]. The work of MJ is supported by the National Institute for Health Research (NIHR) Oxford Biomedical Research Centre (BRC). LL's work is supported by the following grants: R01MH118534, P50MH100029, R01MH118285. Acquisition of the chimpanzee data was funded in part by the Yerkes National Primate Research Center Grant No. ORIP/OD P51OD011132; the Yerkes National Primate Center is supported by the National Institutes of Health, Office of Research Infrastructure Programs/OD [P51OD011132]; the National Chimpanzee Brain Resource is supported by NIH - National Institute of Neurological Disorders and Stroke. KK's work is supported by the BBSRC UK [BB/H016902/1], a

Wellcome Trust Strategic Award [101092/Z/13/Z], and a Heisenberg-Professorship by the Deutsche Forschungsgemeinschaft DFG [KR 5138/1–1].

## Additional information

### Competing interests
Kristine Krug: Reviewing editor, *eLife*. The other authors declare that no competing interests exist.

### Funding

| Funder | Grant reference number | Author |
|---|---|---|
| Wellcome | 203730/Z/16/Z | Nicole Eichert |
| Wellcome | 203139/Z/16/Z | Rogier B Mars |
| National Institute for Health Research | Oxford Biomedical Research Centre | Rogier B Mars<br>Mark Jenkinson |
| Biotechnology and Biological Sciences Research Council | BB/N019814/1 | Rogier B Mars |
| Dutch National Science Foundation | 452-13-015 | Rogier B Mars |
| Academy of Medical Sciences | | Emma C Robinson |
| British Heart Foundation | | Emma C Robinson |
| UK Government | Department of Business, Energy and Industrial Strategy | Emma C Robinson |
| Wellcome | SBF003\1116 | Emma C Robinson |
| European Commission | Marie Sklodowska-Curie Fellowship: 750026 | Katherine L Bryant |
| Medical Research Council | MR/L009013/1 | Saad Jbabdi |
| National Institute for Health Research | | Mark Jenkinson |
| Biotechnology and Biological Sciences Research Council | BB/H016902/1 | Kristine Krug |
| Wellcome | 101092/Z/13/Z | Kristine Krug |
| National Institutes of Health | R01MH118534 | Longchuan Li |
| National Institutes of Health | P50MH100029 | Longchuan Li |
| National Institutes of Health | R01MH118285 | Longchuan Li |

The funders had no role in study design, data collection and interpretation, or the decision to submit the work for publication.

### Author contributions
Nicole Eichert, Conceptualization, Resources, Formal analysis, Funding acquisition, Visualization, Methodology, Writing - original draft, Writing - review and editing; Emma C Robinson, Software, Methodology, Writing - review and editing; Katherine L Bryant, Kristine Krug, Kate E Watkins, Resources, Writing - review and editing; Saad Jbabdi, Mark Jenkinson, Methodology, Writing - review and editing; Longchuan Li, Resources; Rogier B Mars, Conceptualization, Resources, Software, Supervision, Funding acquisition, Methodology, Writing - original draft, Project administration, Writing - review and editing

### Author ORCIDs
Nicole Eichert (ID) https://orcid.org/0000-0001-7818-5787
Katherine L Bryant (ID) https://orcid.org/0000-0003-1045-4543

Saad Jbabdi [ID] https://orcid.org/0000-0003-3234-5639
Mark Jenkinson [ID] https://orcid.org/0000-0001-6043-0166
Longchuan Li [ID] https://orcid.org/0000-0002-0559-0754
Kristine Krug [ID] https://orcid.org/0000-0001-7119-9350
Kate E Watkins [ID] https://orcid.org/0000-0002-2621-482X
Rogier B Mars [ID] https://orcid.org/0000-0001-6302-8631

### Ethics

Human subjects: The study was approved by the Central University (of Oxford) Research Ethics Committee (CUREC, R55787/RE001) in accordance with the regulatory standards of the Code of Ethics of the World Medical Association (Declaration of Helsinki). All participants gave informed consent to their participation and were monetarily compensated for their participation.

Animal experimentation: Chimpanzee data: Data were acquired at the Yerkes National Primate Research Center (YNPRC) at Emory University through separate studies covered by animal research protocols approved by YNPRC and the Emory University Institutional Animal Care and Use Committee (approval no. YER-2001206). These chimpanzee MRI scans were obtained from a data archive of scans obtained prior to the 2015 implementation of U.S. Fish and Wildlife Service and National Institutes of Health regulations governing research with chimpanzees. All the scans reported in this publication were completed by the end of 2012. Macaque Data: Procedures of the in vivo macaque data acquisition were carried out in accordance with Home Office (UK) Regulations and European Union guidelines (EU directive 86/609/EEC; EU Directive 2010/63/EU).

### Decision letter and Author response

Decision letter https://doi.org/10.7554/eLife.53232.sa1
Author response https://doi.org/10.7554/eLife.53232.sa2

## Additional files

### Supplementary files

• Supplementary file 1. Supplementary table related to methods. MSM configuration parameters for the registration of individual subject myelin maps prior to averaging to create a species myelin map. The parameters were kept constant for the three species and for both hemispheres.

• Supplementary file 2. Supplementary table related to methods 'myelin-based surface registration'. MSM configuration parameters. Settings for the MSM command for registrations using the macaque data as input (left panel) and chimpanzee data as input (right panel). The described 'step' refers to the numbering in *Appendix 1—figure 1A*. Parameters were identical for the left and right hemisphere.

• Transparent reporting form

### Data availability

This study used previously published datasets and availability of source data for the different datasets is provided in an overview table in the main manuscript ('Data Availability Overview'). The anonymised human MRI dataset that was generated for the present study is available via OpenNeuro under the accession code ds002634 (version 1.0.1). Result scene files are openly available from the Wellcome Centre for Integrative Neuroimaging's GitLab at https://git.fmrib.ox.ac.uk/neichert/project_MSM (copy archived at https://github.com/elifesciences-publications/project_msm). Group-level myelin-maps and tract surface maps of the three species are openly accessible as part of the result scene files. Numerical data underlying Figure 5 and Appendix 3—figure 2 are provided as source data with the article. All further derived data supporting the findings of this study are available from the corresponding author upon reasonable request.

The following dataset was generated:

| Author(s) | Year | Dataset title | Dataset URL | Database and Identifier |
|---|---|---|---|---|
| Eichert N, Mars RB, Watkins KE | 2020 | Project_larynx | http://dx.doi.org/10.18112/openneuro.ds002634.v1.0.1 | OpenNeuro, 10.18112/openneuro.ds002634.v1.0.1 |

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

## Appendix 1

# Myelin-based surface registration

We implemented a cross-species registration based on surface myelin maps using multimodal surface matching (MSM, *Robinson et al., 2018*; *Appendix 1—figure 1A*). MSM is a cortical surface registration algorithm, which works by projecting sheet-like models of the cortical surface to spheres; then aligning these by driving deformation of an input (or moving) mesh until features on the surface (i.e. myelin map intensities) increase in similarity with those represented on a fixed (or reference) mesh. We used a version of MSM that uses higher-order smoothness constraints and strain-based regularization for the regularization mode. MSM can work with multidimensional feature maps and use cross correlation across features to drive a registration. Multiple registration steps can be combined by using the output of a previous registration as initialization

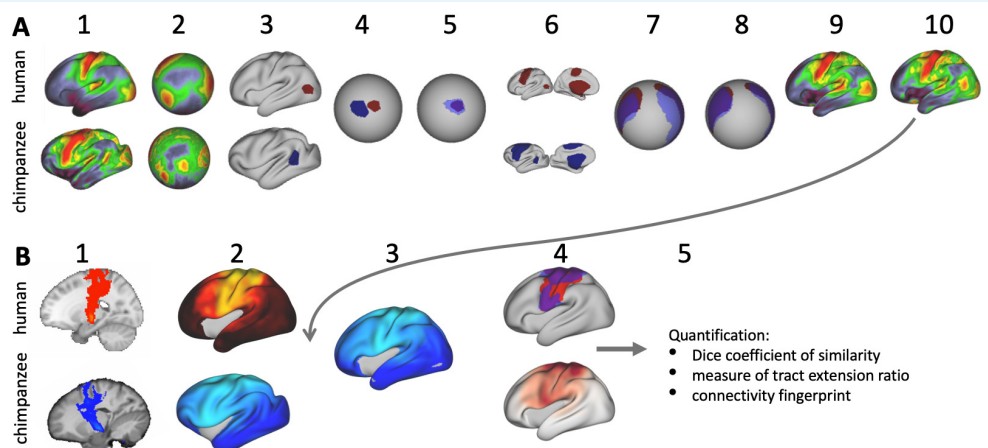

**Appendix 1—figure 1.** Related to Methods. (**A**) MSM registration between chimpanzee and human myelin maps. (1) average species myelin maps. (2) myelin maps displayed on common sphere. (3) ROI for MT+ complex drawn on native surface. (4) MT+ ROI displayed on common sphere. (5) chimpanzee MT+ ROI (blue) registered to human (red) using MSM. (6) Set of three ROIs (MC, MT+, MW). (7) Effect of MT+ ROI-derived initialization on the set of ROIs (only MW and MC visible). (8) Registration of chimpanzee set of ROIs to human ROIs using MSM. (9) Effect of ROI-derived registration on whole hemisphere chimpanzee myelin map. (10) Registration of whole brain myelin maps initialized by set of ROIs using MSM. (**B**) Tract surface analysis. (1) Tractography result for an example tract (CST). (2) Tract map obtained by matrix multiplication. (3) The myelin-derived cross-species registration (A(10)) is applied to transform the actual chimpanzee tract map to human space. (4) Having both tract maps in the same space allows a direct species comparison and quantification of the differences (5).

The main inputs of MSM were the average species myelin maps (*Appendix 1—figure 1A (1)*). All surface and metric files were resampled to a regular 20k-vertices mesh (radius of the sphere: 10 cm). For all species, the same sphere was used for resampling, so that the vertices had correspondence across species (*Appendix 1—figure 1A(2)*).

Given the substantial distortions that are required to match myelin landmarks across species, we initialized the myelin registration using a region-of-interest-(ROI)-driven registration. Three binary ROIs were manually drawn in Connectome Workbench's wb_view onto each species' myelin map (*Appendix 1—figure 1A(3)*). The value inside the ROI was 1 and outside 0. Two ROIs captured the highly myelinated precentral motor cortex (MC) and MT + complex and a third ROI covered the medial wall (MW).

The first initialization step aligned a single ROI for MT+ complex between species to facilitate alignment of the remaining two ROIs (*Appendix 1—figure 1A(5)*). Next, the three

ROIs were combined in a multidimensional file, i.e. in a metric file that contained three data-arrays, or columns (*Appendix 1—figure 1A(6)*). MSM alignment of these sets of ROIs was initialized by the MT+ ROI registration (*Appendix 1—figure 1A(7)*). Initialization was performed using the '–trans=X.sphere.reg.surf.gii'-setting in the MSM-command. In the following step, the whole-hemisphere myelin maps were aligned by using the three-ROI registration step as initialization (*Appendix 1—figure 1A(10)*).

To derive a macaque-to-human registration, we resampled the macaque myelin map to chimpanzee space using the MSM-derived macaque-to-chimpanzee registration. Then we registered the resampled macaque map to the human map while using the chimpanzee-to-human registration as initialization. This approach allowed refinement of the macaque-to-human mapping rather than just applying the chimpanzee-to-human registration to the resampled map. The surface registration was derived for both hemispheres separately with mirrored versions of the ROIs. For registration steps involving ROIs, the registration sphere derived from the left hemisphere was flipped to the right hemisphere. For the steps using myelin maps, we derived the registration separately for both hemispheres. Configuration parameters for all MSM-steps were determined empirically and were kept constant for the two hemispheres (a table with all parameters is provided in *Supplementary file 2*).

## Appendix 2

### Spatial correlation of myelin maps

The quality of the registration performance was visualized by computing a local spatial correlation between the human myelin map and the result of the chimpanzee and macaque registration. The correlation was computed using a sliding window around every vertex on the sphere using MATLAB's corrcoef-function ( Mathworks, Natick, MA). We used a search kernel of 40° that corresponds to a circular search window with a radius of approximately 7 cm.

The vast majority of the cortical myelin maps correlates well after applying the registration (*Appendix 2—figure 1A*). The correlation map shows that inferior frontal parts of the cortex and anterior temporal lobe remain most dissimilar. Despite the good alignment of critical posterior landmarks, such as V1 and MT+ complex, the correlation in posterior parietal areas is lower than in more central areas indicating some residual dissimilarity in the registered myelin maps.

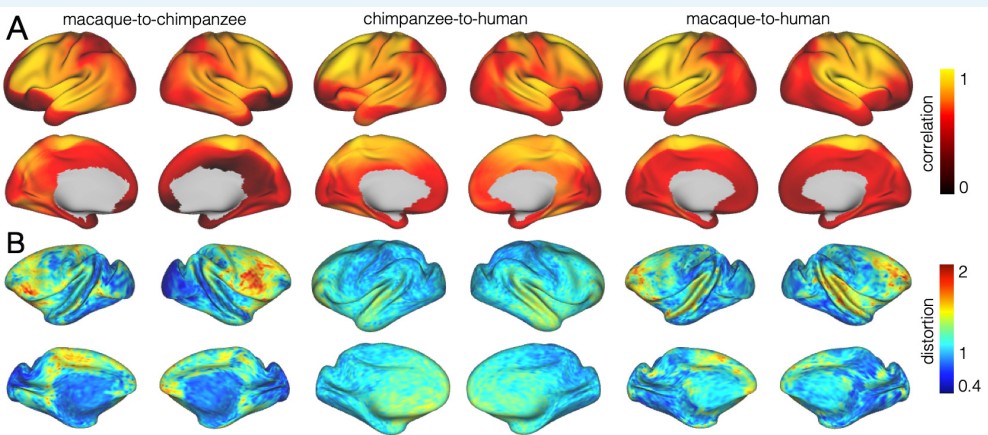

**Appendix 2—figure 1.** Related to *Figure 2*. Myelin correlation and mesh distortion. (**A**) Local correlation of myelin maps after applying the MSM-derived registration. (**B**) Relative distortion of the mesh underlying the registrations. The values indicate a relative expansion (>1) or contraction (<1) of the mesh.

### Expansion maps

To visualize the shifts underlying the three cross-species registrations, we derived a map of surface distortions. We computed the areal distortion between original and distorted mesh, as implemented in 'wb_command -surface-distortion'. Here, distortions for each mesh face are estimated as $\log_2(A_1/A_0)$, where $A_0$ is the area of the original mesh face and $A_1$ is the area of a deformed mesh face. The values per vertex are calculated as a weighted average, where weights are calculated from the relative size of the vertices adjoining mesh faces. We are representing distortions between original and distorted sphere thus the numerical values of the distortion map indicate relative expansion or contraction.

These expansions are underlying the shifts that lead to relocation of cortical areas. A value >1 indicates a relative increase of the underlying mesh triangles and a value <1 indicates a relative decrease in size. Note that the overall increase in brain size across species is not accounted for in this calculation so that some areas show a distortion value smaller than 1.

The macaque-to-chimpanzee distortion map demonstrates that the largest expansion happened in frontal areas (*Appendix 2—figure 1B*). Other areas that show expansion are parietal cortex, posterior temporal and dorsal medial areas. The largest expansion from chimpanzee to human happened in superior temporal and prefrontal cortex, but the distortions are overall smaller and less extended than from macaque to chimpanzee. The

macaque-to-human distortion map shows a similar pattern than chimpanzee-to-human, but indicates a stronger distortion in frontal areas.

## Appendix 3

### Tract maps

In addition to studying tracts in temporo-parietal cortex, we also assessed if the terminations of two further white matter tracts can be predicted by the myelin-based registration. We tracked cortico-spinal tract (CST) and vertical occipital fasciculus (VOF), which both terminate in areas of the cortex that are characteristically high in myelin (*Dum and Strick, 1991*; *Glasser et al., 2014*; *Schmahmann and Pandya, 2009*; *Takemura et al., 2017*). We included these tracts to demonstrate that the registration captures major relocations of highly myelinated areas across the whole cortex. Our proposed framework is thus not restricted to studying temporal lobe architecture, but it can be applied to comparative questions across the whole brain.

In the case of CST, the actual tract projections in the three species show terminations in pre-and post-central gyrus (*Appendix 3—figure 1A,C,E*). For the chimpanzee and macaque, these terminations span a relatively large portion of the frontal cortex, while in the human the anterior part of the frontal lobe does not show any tract terminations. The frontal difference in the three actual tract maps is related to the relative increase in human prefrontal cortex, which is low in myelin content. After applying the myelin-based registration to the macaque and chimpanzee tract maps of CST, the predicted terminations in human space show strong overlap with the actual human tract map. The myelin-based registration thus can model the effect of an increased prefrontal cortex on CST tract terminations. This overlap is captured in the weighted correlation maps, which have a high value in areas surrounding the central gyrus. The Dice coefficients for chimpanzee and macaque CST are relatively high. The extension ratio is close to one in the left hemisphere and slightly elevated for the macaque right hemisphere.

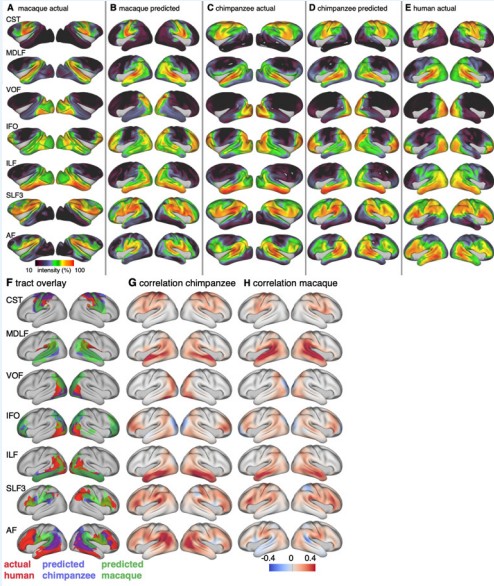

**Appendix 3—figure 1.** Related to *Figure 3* and *Figure 4*. Actual and predicted tract maps and species comparison for the complete set of tracts investigated. Actual tract maps of macaque (**A**), chimpanzee (**C**) and human (**E**). **B** and (**D**) show the transformed tract maps in human space, predicted by the myelin-based registration for macaque and chimpanzee. (**F**): Thresholded actual human tract maps (red) and tract maps predicted by the myelin-based registration for chimpanzee (blue) and macaque (green) (thresholds (t): CST: t = 0.6, MDLF: t = 0.7, VOF: t = 0.6, IFO: t = 0.75, ILF: t = 0.7, SLF3: t = 0.85, AF: t = 0.75). (**G**), (**H**): Weighted correlation maps of actual human map and predicted chimpanzee and macaque map.

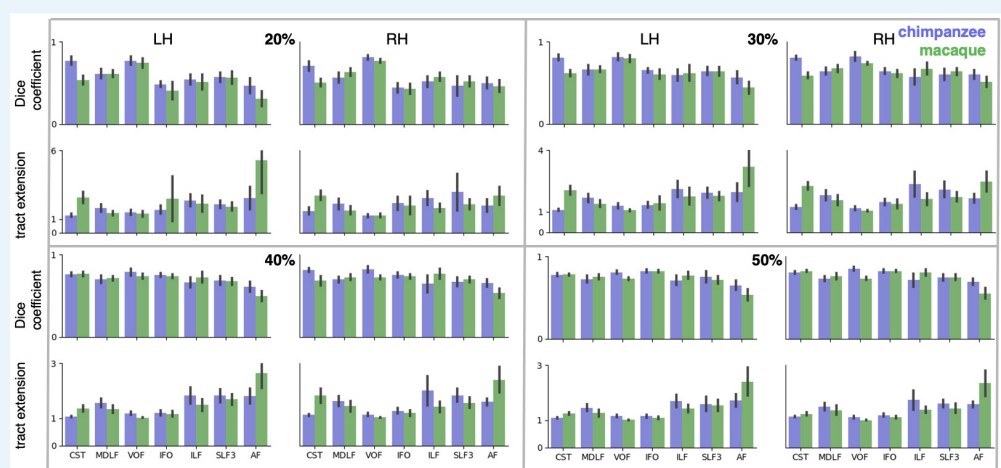

**Appendix 3—figure 2.** Related to *Figure 5*. Dice coefficients and tract extension ratios for different percentages of surface coverage. Shown are mean ± 95% confidence interval derived from all pairs of human (n = 20) and macaque or chimpanzee (n = 5) subjects in left (LH) and right (RH) hemisphere. The low dice coefficient and high extension ratio for AF in macaque and – to a lesser degree – in chimpanzee, is present for all percentages of surface coverage.

The online version of this article includes the following source data is available for figure 2:

**Appendix 3—figure 2—source data 1.** Numerical data underlying the quantifications shown in *Figure 5A,B* and in *Appendix 3—figure 2*.

The posterior terminations of VOF are also predicted well by the myelin-based registration. The predicted cortical terminations of the tracts show a strong overlap with the actual human tract terminations. The termination zones of VOF are found in occipital lobe in all three species. Primary visual cortex has moved from the lateral surface in the macaque to the medial surface in the human, which is modelled well in our myelin registration. When applying the registration, macaque and chimpanzee VOF is predicted to reach posterior and medial parts of the occipital lobe, overlapping well with the actual human tract map. Dice coefficients for VOF are high in both species and the extension ratio close to 1. This result suggests that expansion and relocation of brain areas is largely sufficient to model the posterior cortical terminations of VOF, similar as for MDLF, IFO, ILF and SLF3.

## Appendix 4

# Correlation of tract extension with potential confounds

Since our analyses are based on comparing connectivity maps across species, it is important to show that the differences in connectivity profiles are not due to differences in our *ability* to estimate those connections, but are instead genuine differences in connectivity. We have identified three such sources of confounds: (i) biases in the cortical termination of connections due to differences in gyrification, (ii) noise in the estimate of local fiber orientations due to differences in DW-MRI acquisition, (iii) differences in the geometry of the white matter tracts affecting the correct estimation of their trajectories.

i.   Surface gyrification has been shown to influence tractography results, as streamlines are typically more likely to be detected on the crown of a gyrus than along its banks when tracking from deep white matter towards the cortex (*Jbabdi and Johansen-Berg, 2011*; *Reveley et al., 2015*). We quantified the mean curvature in the surface area covered by the thresholded tract map in all three species. Surface curvature maps were provided by the HCP-pipeline. We then derived the species ratios of curvature for human vs. chimpanzee and for human vs. macaque. We derived a tract-wise correlation between the tract extension ratios (as shown in *Appendix 3—figure 2*) and the curvature ratios.

ii.  Another aspect that affects the performance of the tractography algorithm is the ability to estimate crossing fibers in FSL's bedpostX, which is influenced by differences in the DW-MRI sequence (*Maier-Hein et al., 2017*). This can be quantified by looking at the width of the posterior distribution of the orientation parameters, which in FSL is quantified by the dispersion of the fiber orientations (dyads_dispersion). To test if differences in acquisition parameters affected the dispersion in a way that biased our results, we performed a similar analysis as described above for surface curvature. For each tract we masked the dyad dispersion vector images in the ROI covered by the volumetric tractogram thresholded at 30% and computed the mean value across all three voxel-wise fiber orientations. We then derived a pair-wise species ratio between human vs. chimpanzee (and vs. macaque) and correlated this dispersion ratio with each tract's extension ratio.

iii. The geometry of the tract itself can affect how well the algorithm can reconstruct the pathway. For example, it may be that tracts going through a narrow funnel are more likely to result in false positives/false negatives due to ambiguities between crossing and kissing fibers (*Basser et al., 2000*; *Jones, 2010*; *Seunarine and Alexander, 2013*). We quantify this aspect of the tracts' geometry by estimating the (cosine transformed) angle between crossing fibers in each voxel:

$$divergence_{i,j} = 1 - \cos\left(\left(dyad_i, \, dyad_j\right)\right)$$

This divergence measure was averaged across the three pairs of dyads (where $i$, and $j$ indicate fiber direction) and then averaged within each tract's volumetric extent.

  In addition to the factors described above, which quantify our *ability* to estimate the connectivity, we also evaluated if the quality of the myelin-driven registration biased the tract extension ratios. A spatial map of registration error was derived based on the inverse of the spatial correlation maps of actual human myelin and predicted myelin map (the maps shown in *Appendix 2—figure 1A*), i.e. one *minus* the intensity in the spatial correlation map. In other words, this measure indicated how badly the predicted myelin map based on surface registration correlated with the actual myelin map. For each tract, we then computed the mean of the error map in the area covered by the tract map and we then determined the tract-wise ratio in registration error between species. For the arcuate fasciculus (AF), and the middle longitudinal fasciculus (MDLF) we performed an additional vertex-wise analysis on the relation of registration error and species difference. We calculated a correlation of the vertex-wise value in the registration error map and the absolute difference in the intensity of actual human and predicted tract maps. Finally, we assessed if spatial differences in the myelin signal

affected the results. This was done by quantifying the mean myelin content in the area covered by the tract maps similar as described above for surface curvature.

The tract-wise correlation analyses show that there is no linear relationship between any of the potential confounds and tract extension (r < 0.2, ns) (*Appendix 4—figure 1A*). It can also be observed, that AF predicted by the macaque, which has the strongest tract extension ratio, is not an outlier within the distribution of tracts for any of the measures that we assessed. An exception is the 'complexity' measure, where AF is found at the lower end of the distribution. This indicates merely that AF differs in the complexity measure compared to the other tracts, but this does not indicate that the effect in tract extension is driven by tract complexity. Other tracts that have a high extension ratio (e.g. chimpanzee ILF in the right hemisphere) are found at the other end of the distribution.

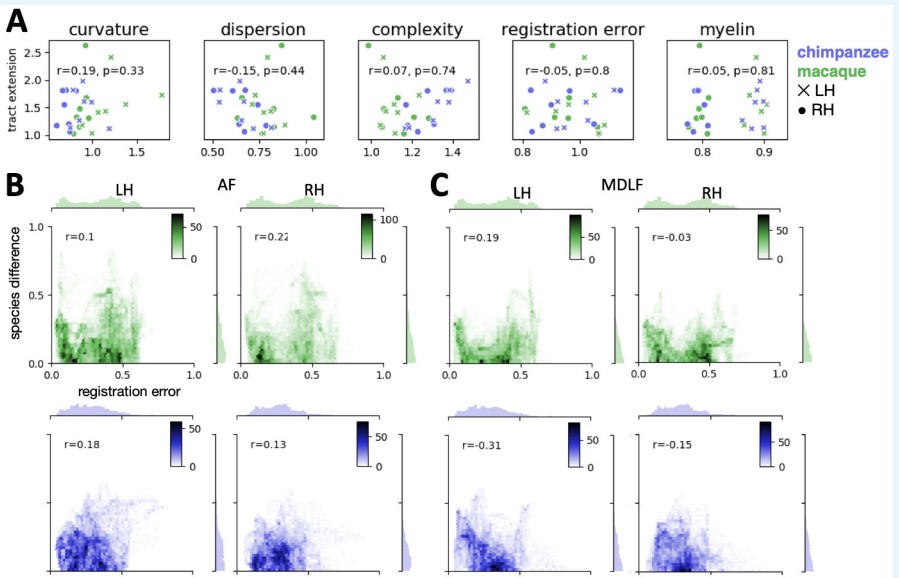

**Appendix 4—figure 1.** Related to *Figure 5* and *Appendix 3—figure 2*. (**A**) Tract-wise correlation of tract extension ratio and potentially confounding factors (n = 28). Labels for tracts are not provided for readability, but can be inferred from *Appendix 3—figure 2*. For details refer to the text. (**B, C**) Whole-brain vertex-wise correlation of absolute species difference in actual and predicted tract map (**B**: AF; **C**: MDLF) and the registration error map (i.e. one minus the myelin correlation map in *Appendix 2—figure 1*). Marginal histograms show the distribution of vertices across 50 regular bins. The inset scale shows the number of vertices within the bins of the joint distribution scatterplot (color mapping as in **A**), n = 20252 vertices, p<0.001 for all correlations in **B** and **C**.

In addition to the tract-wise analyses, we also performed a vertex-wise analysis on the whole-brain tract map of AF (*Appendix 4 – Figure 1B*). A correlation between registration error and absolute species difference in tract map intensities (actual human *minus* predicted) shows that there is no linear relationship for both hemispheres and both species (r < 0.22, p<0.001). For visual comparison, we plotted the 2D joint distribution of registration error and species difference for AF, where we observed the strongest tract expansion, and for MDLF, where we observed only small tract expansion (*Appendix 4 – Figure 1B, C*). The joint distribution plots of the two tracts show a similar pattern, which demonstrates that species differences in AF cannot be attributed to an unusual bias due to registration error.

In sum, these control measure show that our observed effects of tract extension cannot be explained by potentially confounding factors, such as gyral bias and registration error.

