## [Decision Letter]

Thank you for submitting your article "Cross-species cortical alignment identifies different types of anatomical reorganization in the primate temporal lobe." for consideration by *eLife*. Your article has been reviewed by three peer reviewers, including Timothy Verstynen as the Reviewing Editor and Reviewer #1, and the evaluation has been overseen by Joshua Gold as the Senior Editor. The following individual involved in review of your submission has agreed to reveal their identity: Katja Heuer (Reviewer #2).

Summary:

This manuscript describes a cross species analysis of temporal lobe organization between three primate species (macaque, chimpanzee, and humans). Using cortical myelin mapping to define regional organization and diffusion weighted imaging (DWI) tractography to define connectivity profiles, the authors tested the hypothesis that species differences in temporal lobe organization do not just reflect expansion/movement of specific modular regions, but also a fundamental reorganization of connection topology as well. The simple null hypothesis is that species-differences in the location of white matter fascicle terminations will be completely explained by migration of cortical regions. The authors found that, for a subset of temporal lobe regions, including the arcuate, ILF, and SLF, the shift in connection topology from macaque-to-chimp-to-humans is not completely explained by migration of cortical areas. The authors conclude that regions undergo different evolutionary modifications beyond the location and extent of distinct functional regions in the cortex.

All three reviewers found this to be a very clear, compelling, and contributes a valuable insight into the anatomy and possible evolutionary scenarios of several major temporo-parietal fiber tracts across 3 primate species, as well as with a refined method for a cross-species registration. The hypothesis is rational and clear, both in what it predicts and how it links to the methods used to evaluate it. The story that emerges is interesting both from an evolutionary neuroscience perspective and for understanding the specialization of the temporal lobe in humans.

Over the course of the review and consultation, a few critical concerns were brought up that fall under common themes. These concerns are consolidated here.

Essential revisions:

1) Hypothesis evaluation.

Reviewer 1 found that the logic of the hypotheses as laid out in Figure 1 are quite clear and follow a rational logic: if differences in white matter topology are simply following the movement of functionally specialized regions across evolution, then accounting for the migration of cortical areas should full explain endpoint locations in the major white matter fascicles that are preserved across species. However, this all relies on the assumption that the measures used to both map cortical regions and map connectivity are veridical and without noise or bias. Unfortunately we know that this is not the case, especially with the DWI tractography. The authors should see these papers for a review of these problems:

– Thomas, C., Frank, Q. Y., Irfanoglu, M. O., Modi, P., Saleem, K. S., Leopold, D. A., and Pierpaoli, C. (2014). Anatomical accuracy of brain connections derived from diffusion MRI tractography is inherently limited. Proceedings of the National Academy of Sciences, 111(46), 16574-16579.

– Maier-Hein et al., 2017.

Not only are DWI tractography results sensitive to biases with respect to resolving individual fascicles depending on their geometry, but differences in DWI sequences can lead to different tractography results. Both are problems for comparing across species: i.e., the same pathway may have different geometries that make it easier or harder to track reliably and each species was imaged using different DWI sequence parameters (e.g., TR, number of directions, diffusion gradient strength).

This leaves an alternative explanation for the results shown in Figures 3-5: perhaps noise in the tractography process leads to different cortical endpoint fields in the different species. Right now, it is impossible to distinguish between this and a reorganized connectivity profile hypothesis. You should find a way to vet the connectivity reorganization hypothesis against the "noisy measures" hypothesis.

Reviewer 3 raised a similar concern, pointing out that if some tracts have extremely low overlap between predicted vs. actual – can this be attributed to real evolutionary/phylogenetic differences or an artefact of worse accuracy in transformation? Reviewer 3 shared one proposal for how this could be quantifiable using existing data and techniques used in the paper. The authors should feel free to improve this suggestion.

If we can create a distribution of how extent of errors in registration map to errors in tractography predictions (dice overlap, predicted tract extension), it would provide an upper bound of the largest possible discrepancies in tractography predictions. Then if the observed non-overlap in AF tracts exceed what one might expect due to registration errors alone, then indeed this provides more definitive evidence that such non-overlap can truly be attributed to phylogenetic distance between species. One way to do this is as follows:

– Step 1: In order to align macaque to human brains the paper suggests that a composition of transforms from macaque to chimp then chimp to human provides more accurate registration than a direct macaque to human mapping. One option to investigate the consequence of registration errors is then to compare the superior but indirect macaque to human transforms with the less accurate direct transform. Ultimately any macaque to human comparison has the same phylogenetic difference, so the discrepancy between the direct and indirectly transformed myelin maps offer a distribution of registration errors.

– Step 2: Analogous to analyses already performed in the paper, one can also perform tractography predictions using the direct macaque to human transform, in addition to the indirect transform already performed. Any discrepancies between the direct and indirect predictions are now attributable to registration errors.

– Step 3: For each of the investigated tracts it would be useful to create a 2D joint distribution of registration discrepancy and tractography discrepancy. This would provide an overall picture of how worse registration might lead to worse tractography predictions and thus provide a useful guideline for follow-up studies.

– Step 4: The bar plots in Figure 5 can be augmented with an additional bar corresponding to the less accurate macaque-human transform, to act as a "secondary control" for the current macaque to human comparison. Unfortunately, I cannot think of a way to provide a similar control for the chimp to human comparison.

2) Concerns with spatial alignment.

Reviewer 2 noted that some cortical regions are very compressed when mapped into a sphere, in particular, the frontal pole or the temporal pole. Have you evaluated the impact that this may have on a cross-species registration? And with larger geometric differences between 2 of the species as compared to the third?

Reviewer 3 asked what factors prevent the macaque/chimp to human projection from being an unbiased one? Suppose that one has an oracle that could learn the theoretically best possible cross-species registration by perfectly mapping all changes due to expansion/relocation all over the brain. Where might we expect myelin based registrations to differ from such an oracle?

a) Accuracy of spatial registration maybe unevenly distributed. The authors allude to this in the Discussion. Appendix 2—figure 1, also provides evidence of mesh distortion. I take these maps to be evidence of potentially uneven accuracy of spatial registration. There certainly seems to be some evidence that frontal areas and temporal areas have non-trivial mesh distortion. These overlap with the areas where the tractography of arcuate fasciculus fail to overlap across species.

b) The authors also demonstrate how surface coverage affects the overlap between predicted and actual tractography in Figure 5. It is certainly evident from this figure that low surface overlap exaggerates the human/non-human discrepancy.

c) While these investigations are extremely useful but only serve to highlight that thoroughly accounting for the effect of registration errors are important. They don't cover the possibility that the areas where species actually differ and likely more prone to registration errors and thus might exaggerate or compound the changes in tract length attributable to evolution.

2) Myelin mapping. The primary measure of distinct cortical regions is the T1w/T2w ratio maps thought to reflect differences in cortical myelin. As is shown in Figure 1C (and Figure 2), these maps are largely biased towards primary cortical regions (both sensory and motor). Yet a vast majority of the temporal lobe is association cortex. How do we know that there is enough reliable myelin signal in the temporal association areas to know that the across-species alignment is accurate? How similar does the mapping look when using another measure? Could this bias towards primary regions (e.g., A1) explain why some tracts are better aligned than others?

3) Connectivity fingerprints.

Reviewer 1 was confused as to what the connectivity fingerprints are showing. Even after digging into the Materials and methods, they are still not entirely sure what they are or how the interpretations being made map to the data presented. For example, what would a null finding really look like in these results (Figure 5)? A lot more detail needs to be provided, both in the Results and Materials and methods to clarify what these are and how they can be interpreted within the context of the paper.

Reviewer 2 had a similar concern, pointing out that connectivity data and myelin maps are not fully independent features, though, but could be considered rather the one conditioning the other. In particular, from a developmental point of view, connections need to be formed before they can be myelinated. What gives pre-eminence to one modality over another?

Reviewer 3 was concerned about the nature of the alignment used to evaluate the fingerprints, pointing out that, in the subsection “Predicted Tract Maps”, the authors explain that myelin based spatial transforms are applied to the derivative tract maps after doing DWI tractography in the species-native space. Why not apply registration directly to the DWI images first and then conduct tractography itself in the human-aligned space rather than applying to derived maps? Given the novelty of this paper, it would be useful to make clear if this is a potential avenue for methodological interest. On the other hand, if there is a serious flaw with the approach I propose, it would be also useful to clarify this. I don't see any discussion of this choice.

4) Data sharing.

Reviewer 2 raised an issue of data availability as a means of expanding the impact of the current study. If it were possible, the authors may consider sharing their data more openly (not only upon request), and also sharing raw data to improve the replicability of their findings and impact on the community. The code instructions inside their shared script folder ensure reusability of the method by the community.

Furthermore, the authors may consider adding a note on the sharing status of the original data they use to their manuscript. For example, the data from the National Chimpanzee Resource, is made available upon request to W. Hopkins. Including such information would help the community and encourage re-use of valuable data resources (or to not lose their time trying to track data sources for possible re-use).

In-vivo structural and DWI Chimp data obtained from W. Hopkins, NCBR – available upon request with W. Hopkins.

Myelin maps of 29 chimps (Donahue and Glasser data) – available?

DWI data for the 5 chimps (Mars) – available?

Macaque ex-vivo structural MRI data (Mars) – available?

1 ex-vivo macaque (de Crespigny) – available?

T1w/T2w myelin maps macaque (Mars?) – available?

---

## [Author Response]

Essential revisions:1) Hypothesis evaluation.Reviewer 1 found that the logic of the hypotheses as laid out in Figure 1 are quite clear and follow a rational logic: if differences in white matter topology are simply following the movement of functionally specialized regions across evolution, then accounting for the migration of cortical areas should full explain endpoint locations in the major white matter fascicles that are preserved across species. However, this all relies on the assumption that the measures used to both map cortical regions and map connectivity are veridical and without noise or bias. Unfortunately, we know that this is not the case, especially with the DWI tractography. The authors should see these papers for a review of these problems:– Thomas, C., Frank, Q. Y., Irfanoglu, M. O., Modi, P., Saleem, K. S., Leopold, D. A., and Pierpaoli, C. (2014). Anatomical accuracy of brain connections derived from diffusion MRI tractography is inherently limited. Proceedings of the National Academy of Sciences, 111(46), 16574-16579.– Maier-Hein et al., 2017.

We want to thank the reviewers for their comments and their suggestions on our manuscript. The reviewers raised concerns regarding potentially confounding factors that might have influenced the results. In the following we will discuss how we addressed these concerns.

Reviewer 1 emphasizes the challenges in diffusion-based tractography, in particular concerning differences in the endpoint of tracts that could explain their observed differences across tracts. We addressed this concern by using an approach that mitigates the influence of endpoint morphology: To derive the surface tract maps, we multiplied each tract’s connectivity matrix in volume space with a vertex-to-whole brain connectivity matrix (Mars et al., 2018b). This method has previously shown to reduce gyral bias when compared to surface-based tractography approaches (Eichert et al., 2018). As a further refinement of this approach, we weighted the connectivity matrix based on the voxel-wise vertex-to-voxel distances, which resulted in more evenly distributed surface tract maps, which are better comparable across species.

Moreover, to ensure that differences in cortical geometry are not driving the observed results, in our revised manuscript we now also assessed if the surface curvature, a measure of sulcal architecture, underlying each tract map correlates with tract expansion ratio, which is our main outcome measure (see Appendix 4—figure 1). If gyral bias, which is one of the problems highlighted by the papers indicated by the reviewer, is a confounding factor, we should observe a strong relationship between the sulcal architecture of a tract and its expansion ratio. No linear relationship was observed (*r* = 0.19, *ns*) and the macaque arcuate fasciculus (AF), where we found the strongest tract expansion, is not found to be an outlier within this distribution. This information and information addressing the other issues discussed below, as well as references to the papers mentioned by the reviewer and the additional figure, has now been added to the manuscript:

Methods section:

“We performed additional control analyses to assess if the observed effects of tract expansion correlate with potential sources of confounds arising from our connectivity measures and the myelin-driven registration. The methodology and results of these control analyses are reported in Appendix 4.

Appendix 4:

“Correlation of tract extension with potential confounds

Since our analyses are based on comparing connectivity maps across species, it is important to show that the differences in connectivity profiles are not due to differences in our *ability* to estimate those connections, but are instead genuine differences in connectivity. […] In sum, these control measure show that our observed effects of tract extension cannot be explained by potentially confounding factors, such as gyral bias and registration error.”

Not only are DWI tractography results sensitive to biases with respect to resolving individual fascicles depending on their geometry, but differences in DWI sequences can lead to different tractography results. Both are problems for comparing across species: i.e., the same pathway may have different geometries that make it easier or harder to track reliably and each species was imaged using different DWI sequence parameters (e.g., TR, number of directions, diffusion gradient strength).This leaves an alternative explanation for the results shown in Figures 3-5: perhaps noise in the tractography process leads to different cortical endpoint fields in the different species. Right now it is impossible to distinguish between this and a reorganized connectivity profile hypothesis. You should find a way to vet the connectivity reorganization hypothesis against the "noisy measures" hypothesis.

As the reviewer noted, the performance of the tractography algorithm can be influenced by differences in the in DW-MRI acquisition parameters in the different species, and by the geometry of the tracts. To assess if our observed effects are driven by these differences, we performed similar control analyses as described for the reviewer’s comment above. Differences in the posterior distribution of the orientation parameters fitted by FSL’s bedpostX can be quantified using the dispersion of fiber orientations (‘dyad_dispersion’). A tract’s geometry can be estimated by the angle between crossing fibers in each voxel. For each tract in each species we derived a comparable pair-wise measure for dyad dispersion and tract complexity. Then, we derived a correlation between the tract expansion ratio and the dispersion and complexity measures. No linear relationship can be observed (dispersion: *r* = -0.15, *ns*; complexity: *r* = 0.07, *ns*), indicating that these potentially confounding factors cannot explain the observed results.

To describe the additional quality control analyses, we added Appendix 4.

Reviewer 3 raised a similar concern, pointing out that if some tracts have extremely low overlap between predicted vs. actual – can this be attributed to real evolutionary/phylogenetic differences or an artefact of worse accuracy in transformation? Reviewer 3 shared one proposal for how this could be quantifiable using existing data and techniques used in the paper. The authors should feel free to improve this suggestion.If we can create a distribution of how extent of errors in registration map to errors in tractography predictions (dice overlap, predicted tract extension), it would provide an upper bound of the largest possible discrepancies in tractography predictions. Then if the observed non-overlap in AF tracts exceed what one might expect due to registration errors alone, then indeed this provides more definitive evidence that such non-overlap can truly be attributed to phylogenetic distance between species. One way to do this is as follows:– Step 1: In order to align macaque to human brains the paper suggests that a composition of transforms from macaque to chimp then chimp to human provides more accurate registration than a direct macaque to human mapping. One option to investigate the consequence of registration errors is then to compare the superior but indirect macaque to human transforms with the less accurate direct transform. Ultimately any macaque to human comparison has the same phylogenetic difference, so the discrepancy between the direct and indirectly transformed myelin maps offer a distribution of registration errors.– Step 2: Analogous to analyses already performed in the paper, one can also perform tractography predictions using the direct macaque to human transform, in addition to the indirect transform already performed. Any discrepancies between the direct and indirect predictions are now attributable to registration errors.– Step 3: For each of the investigated tracts it would be useful to create a 2D joint distribution of registration discrepancy and tractography discrepancy. This would provide an overall picture of how worse registration might lead to worse tractography predictions and thus provide a useful guideline for follow-up studies.– Step 4: The bar plots in Figure 5 can be augmented with an additional bar corresponding to the less accurate macaque-human transform, to act as a "secondary control" for the current macaque to human comparison. Unfortunately, I cannot think of a way to provide a similar control for the chimp to human comparison.

We thank reviewer 3 for their comments and for the valuable suggestion for a quantification. To show that our effects of tract expansion are not driven by registration error, we sought to implement the approach that the reviewer proposed. However, due to the large shifts required, a direct registration from macaque to human is technically difficult to obtain using MSM. This is why we introduced an intermediate resampling step to chimpanzee space. Moreover, if we implemented the approach, we would not be able to obtain a similar control measure for the chimpanzee-to-human registration that would allow us to compare the effects across species. It should be noted that it is conceptually difficult to fully validate registration errors as there is no absolute ground truth for the shared coordinate system, which is also a problem for human-to-human registration.

Therefore, to implement the reviewer’s suggestion to investigate whether the quality of registration could explain our observed effects, we assessed the effect of registration error on species differences using the spatial correlation maps of predicted and actual human myelin map (see Appendix 2—figure 1A). These myelin correlation maps indicate the accuracy of the myelin alignment across species. We tested if the tracts’ expansion ratios (i.e. the pair-wise measure of how much we judged a tract to be expanded between species) correlate with the registration error (i.e. 1 *minus* accuracy), similar as described above for surface curvature. No linear relationship between mean registration error and tract expansion ratio was observed (*r* = 0.05, *ns*).

In addition to the tract-based correlation analysis, we performed a vertex-wise analysis for the arcuate fasciculus (AF), the tract where we observed the strongest tract expansion (new figure: Appendix 4—figure 1). If a species difference in AF was an artefact of worse registration accuracy, we would expect a relationship between species difference and registration error. A vertex-wise correlation of absolute difference in intensity values of the actual and predicted tract maps and registration error revealed no linear relationship (*r* < 0.22, for both macaque and chimpanzee in both hemispheres). As the reviewer suggested, as visualization we plotted the 2D joint distribution maps for species difference and registration error. We also performed a vertex-wise analysis for a tract where we did not find strong species differences, the middle longitudinal fasciculus (MDLF). The joint distribution maps of AF and MDLF show overall a very similar pattern. This demonstrates that the registration error is not biasing species differences in a way that would bias us to find an effect for AF.

To describe the additional quality control analyses, we added Appendix 4, as shown above.

2) Concerns with spatial alignment.Reviewer 2 noted that some cortical regions are very compressed when mapped into a sphere, in particular, the frontal pole or the temporal pole. Have you evaluated the impact that this may have on a cross-species registration? And with larger geometric differences between 2 of the species as compared to the third?

We thank the reviewer for their suggestion. Using anatomical surface information in the species registration will certainly make a promising avenue future studies. The original MSM method is a spherical registration approach, which simplifies the problem of cortical registration by projecting a convoluted surface to a sphere. As the reviewer pointed out, a limitation of this framework is that a projection to the sphere distorts the relative distance of vertices between the original anatomical surfaces. For this reason, a modified version of MSM has been proposed, which enables regularization of deformations based on the anatomical surface (aMSM, Robinson et al., 2018). The computational load of performing an anatomical registration, however, is significantly higher. Whilst we have tried running aMSM, the default parameters (configured for neonatal cortical development) are not appropriate for this task. Re-optimization would present a significant computational bottleneck and would benefit from specialist expertise (in the properties of biomechanical strain for cross species deformations) that is not presently available. Given these constraints we plan to pursue this as a future, separate extension study in our laboratory.

Reviewer 3 asked what factors prevent the macaque/chimp to human projection from being an unbiased one? Suppose that one has an oracle that could learn the theoretically best possible cross-species registration by perfectly mapping all changes due to expansion/relocation all over the brain. Where might we expect myelin based registrations to differ from such an oracle?

The notion of an oracle that could learn a perfect cross-species registration is misleading, because even in theory such a perfect registration does not exist, since different species do after all have different brains. All aspects of brain architecture can be modified during the course evolution and each individual aspect would provide us with a unique registration. Our aim in this study is not to find the ‘best’ species registration, but to shed light on brain evolution by studying where registrations based on different modalities disagree.

We used myelin maps as an index of how cortical areas expanded and relocated over the brain. This is a heuristic, but it is a measure that is available in all three primate species. An alternative would have been to derive a registration based on cytoarchitectonic maps. We expect that this registration would provide us with more defining features in the frontal lobe, where myelin features are low. However, homologous cytoarchitectonic maps are difficult to obtain across multiple species. We have used the myelin map here as a type of map readily available and one that can potentially be obtained in future studies in other species (e.g. marmoset).

To address the reviewer’s point, we have included the following statement in the Discussion section:

“The aim of our research is not to find the ‘best’ species registration, but to shed light on brain evolution by studying where registrations based on different modalities disagree. […] The crucial point is that the maps we employed here are similar across species, allowing us to compare like with like (Glasser et al., 2014).“

a) Accuracy of spatial registration maybe unevenly distributed. The authors allude to this in the Discussion. The Appendix 2—figure 1, also provide evidence of mesh distortion. I take these maps to be evidence of potentially uneven accuracy of spatial registration. There certainly seems to be some evidence that frontal areas and temporal areas have non-trivial mesh distortion. These overlap with the areas where the tractography of arcuate fasciculus fail to overlap across species.

As described in our response above, we critically assessed the effect of registration error on tract expansion. The distortion maps in Appendix 2—figure 1B show which parts of cortex relatively increased or decreased in size during the myelin registration. These maps do not indicate the spatial registration accuracy. The term distortion is used here, rather than expansion, because it describes the strength of the regularization. An expansion, i.e. distortion, in frontal and temporal areas is expected and in line with previous literature (Glasser et al., 2014, Hill et al., 2010), and it does not provide evidence for aberrant mesh distortions.

b) The authors also demonstrate how surface coverage affects the overlap between predicted and actual tractography in Figure 5. It is certainly evident from this figure that low surface overlap exaggerates the human/non-human discrepancy.

Our outcome measures in Figure 5 and Appendix 3—figure 2 (dice coefficient of overlap and tract expansion) are derived from thresholded tract maps. As each tract has a different distribution of intensity values on the surface, we derived the threshold based on percentage of surface coverage rather than intensity value of the tract map. The same threshold was applied for each species. A percentage coverage of 40% was empirically found to provide the most robust effect. In Appendix 3 we also show data for a surface coverage of 20%, 30% and 50%. As the reviewer noted, the observed tract expansion for the left macaque AF is especially high at a surface coverage of 20%, but also the variability across subjects is greatly increased. Overall, Appendix 3—figure 2 shows that the strongest effect can be observed for AF, independent of surface coverage. Our choice of threshold thus did not affect our results.

To highlight this point we added the following sentence to the legend of Appendix 3—figure 2:

“The low dice coefficient and high extension ratio for AF in macaque and – to a lesser degree – in chimpanzee, is present for all percentages of surface coverage.“

c) While these investigations are extremely useful but only serve to highlight that thoroughly accounting for the effect of registration errors are important. They don't cover the possibility that the areas where species actually differ and likely more prone to registration errors and thus might exaggerate or compound the changes in tract length attributable to evolution.

This is an extremely important point. The goal of our approach is NOT to derive the best possible registration between species. It is to derive the best possible registration *based on a particular modality* and gain insight into whether the *other* modalities are well predicted. Thus, it is incorrect to interpret our effects as due to ‘registration errors’. The fact that not all tracts are equally well predicted is a feature, not a bug.

The reviewer is perfectly right that we have to ensure that our results are not driven by confounding factors, and we have added a number of quality control analyses (described above and in the new Appendix 4) that show no indication that the observed tract expansion in AF is driven by uneven accuracy of spatial registration.

2) Myelin mapping. The primary measure of distinct cortical regions is the T1w/T2w ratio maps thought to reflect differences in cortical myelin. As is shown in Figure 1C (and Figure 2), these maps are largely biased towards primary cortical regions (both sensory and motor). Yet a vast majority of the temporal lobe is association cortex. How do we know that there is enough reliable myelin signal in the temporal association areas to know that the across-species alignment is accurate? How similar does the mapping look when using another measure? Could this bias towards primary regions (e.g., A1) explain why some tracts are better aligned than others?

We used T1w/T2w surface maps as feature to drive the cross-species surface registration. These maps have empirically been shown to correlate with cortical myelin content (Glasser et al., 2014), but this is merely a heuristic and the underlying biophysical models are not fully understood. However, it has been shown to be a reliable map that can be obtained from multiple species (e.g., Glasser et al., 2014; Donahue et al., 2018). Due to the nature of the signal, the contrast in myelin features is low in some parts of the cortex, such as the frontal lobe and the anterior temporal lobe. We note that registration is based on differences in myelin across the cortex, so both high and low myelin values contribute to the registration.

To verify that myelin signal did not bias our observed effects, we performed an additional control analysis. We quantified if the species ratio in mean myelin content within the surface area covered by the tract maps correlates with each tract’s extension ratio. The result shows that there is no linear relationship between myelin content and tract extension ratio (*r* = 0.05, *p* = 0.81, *n* = 28). AF, where we observed strongest tract extension, falls within the middle of the distribution of the mean myelin content measure. This analysis thus confirms that myelin signal did not bias the registration towards finding species differences in temporal cortex.

To describe the additional quality control analyses, we added Appendix 4 as shown above.

3) Connectivity fingerprints.Reviewer 1 was confused as to what the connectivity fingerprints are showing. Even after digging into the Materials and methods, they are still not entirely sure what they are or how the interpretations being made map to the data presented. For example, what would a null finding really look like in these results (Figure 5)? A lot more detail needs to be provided, both in the Results and Materials and methods to clarify what these are and how they can be interpreted within the context of the paper.

We thank the reviewer for this critical remark. The connectivity fingerprints were derived using the intensity values of the actual and predicted tract maps at the selected two voxels. In the case of a ‘null finding’, the profile of tract intensities (i.e. the height of the bars in the polar plot) would be highly similar for the actual human and the predicted profile. In the parietal lobe vertex, the profiles match well across species, which is in line with the results reported in the rest of the manuscript. In the temporal lobe vertex strong discrepancies can be observed for AF, which is expected. The connectivity fingerprints thus serve as additional visualization and characterization of the main result that emerged from the species comparison. To clarify this quantification, we added more detail in the Materials and methods and the Results section.

Changes to manuscript: The description of the connectivity fingerprints in the Methods and Results sections was rewritten substantially and we made minor changes to the legend for Figure 5.

Materials and methods section:

“Connectivity Fingerprints

We characterized the effect of cortical expansion on brain connectivity using the concept of connectivity fingerprints (Passingham et al., 2002). […] The whole set of tracts investigated (CST, MDLF, VOF, IFO, ILF, SLF3 and AF) was included to give a more detailed estimate of the connectivity fingerprint.”

Results section:

“Connectivity Fingerprints

To further characterize the effects in the predicted tract maps, we obtained connectivity fingerprints at two representative vertices on the left brain surface: One in inferior parietal lobe, where most tracts are predicted well and one in the middle temporal gyrus, where we observed strong species differences, in particular for AF. […] Thus, the connectivity fingerprints of the two representative areas match the pattern of species differences that emerged from the results above.”

Figure 5 legend:

“Figure 5. Quantification of cross-species comparisons. […] C: Connectivity fingerprints at two vertices in inferior parietal and temporal lobe derived from the intensity values of an extended set of tract maps. Shown are mean and standard deviation (human: *n* = 20, chimpanzee and macaque: *n* = 5).“

Reviewer 2 had a similar concern, pointing out that connectivity data and myelin maps are not fully independent features, though, but could be considered rather the one conditioning the other. In particular, from a developmental point of view, connections need to be formed before they can be myelinated. What gives pre-eminence to one modality over another?

We thank the reviewer for the comment, which we would like to discuss here. In the manuscript, we are not implying that cortical myelin content and brain connectivity are completely independent aspects of brain architecture. Indeed, if no changes in brain organization had occurred through evolution, we would expect the relationship between the two aspects to be identical across all three brains. When comparing these modalities across species, however, it is a reasonable assumption that evolutionary adaptations affected cortical myelin content and tract terminations not equally across the whole brain. We show that for large parts of the brain, the tract terminations follow the spatial rearrangement that is observed for the myelin map, but not in some temporal lobe tracts.

Testing the effect of a myelin registration on connectivity is not meant to imply pre-eminence of myelin over connectivity. The present study represents only one example of how cortical specializations can be studied by comparing cross-species registrations of different modalities. In fact the reversed approach, i.e. to derive a registration based on individual tract maps and then test their differential effect on transformed myelin maps, would be highly informative. And our framework is, of course, not limited to these modalities but could be complemented using other cortical features of interest, as described in the Discussion section. We chose myelin here as the primary feature, because we wanted to test the specific hypothesis that some tracts expanded while others simply followed relocation of areas across the cortex. Thus, we used a measure that could index the relocation (myelin) and test it on our measure of interest (the tracts).

We added a statement to highlight this point in the Discussion section of the revised manuscript, which we referred to in a comment above.

Reviewer 3 was concerned about the nature of the alignment used to evaluate the fingerprints, pointing out that, in the subsection “Predicted Tract Maps”, the authors explain that myelin based spatial transforms are applied to the derivative tract maps after doing DWI tractography in the species-native space. Why not apply registration directly to the DWI images first and then conduct tractography itself in the human-aligned space rather than applying to derived maps? Given the novelty of this paper, it would be useful to make clear if this is a potential avenue for methodological interest. On the other hand, if there is a serious flaw with the approach I propose, it would be also useful to clarify this. I don't see any discussion of this choice.

We thank the reviewer for this suggestion. Registration based on DWI images is an interesting methodological approach, but it comes with some technical limitations.

Firstly, a cross-species registration of DWI images would require some form of initialization based on known homologies. Otherwise, an algorithmic regularisation would simply match the voxels based on intensities, which might lead to biologically implausible alignments. For a similar reason, we initialized the myelin registration using three ROIs, where homology across species can be assumed. Secondly, a non-linear registration of DWI images would involve interpolations and potential data degradation that can affect the tract reconstruction using tractography. This is one reason why we performed tractography in native subject space, which is generally the standard in the field, rather than in transformed standard space. Thirdly, given that this registration would be performed in 3D volumetric rather than in 2D surface space, the degrees of freedom would be greatly increased, so that more priors would be needed to constrain the algorithm.

As alluded in the comment above, however, we think the suggested approach – registering based on connectivity and then applying this registration to myelin – could be adapted for surface space.

On a more general note, we believe that projecting the data of different modalities all to a surface representation is a useful tool for comparative neuroscience. It allows us to visualize the different modalities within the same cortical sheet and to compare topologies on this 2D surface. This opens up a wide array of mathematical tools to use of which MSM, the method employed here, is a prime example. Similar approaches are being taken by other groups, for instance in investigating the relationship between gene expression and myelin content of the cortex (Burt et al., 2018) and gradients of change across multiple modalities of brain organization (Huntenburg et al., 2018, Trends Cogn Sci; Blazquez Freches et al., in press, Brain Struct Funct).

To highlight this point, we added the following statement in the Discussion section:

“Projecting data of different modalities to a surface representation is a useful tool for comparative neuroscience. […] A similar approach has been taken to investigate the relationship between gene expression and myelin content of the cortex (Burt et al., 2018) and gradients of change across multiple modalities of brain organization (Huntenburg et al., 2018, Blazquez Freches et al., in press, Brain Struct Funct).”

4) Data sharing.Reviewer 2 raised an issue of data availability as a means of expanding the impact of the current study. If it were possible, the authors may consider sharing their data more openly (not only upon request), and also sharing raw data to improve the replicability of their findings and impact on the community. The code instructions inside their shared script folder ensure reusability of the method by the community.Furthermore, the authors may consider adding a note on the sharing status of the original data they use to their manuscript. For example, the data from the National Chimpanzee Resource, is made available upon request to W. Hopkins. Including such information would help the community and encourage re-use of valuable data resources (or to not lose their time trying to track data sources for possible re-use).In-vivo structural and DWI Chimp data obtained from W. Hopkins, NCBR, – available upon request with W. Hopkins.Myelin maps of 29 chimps (Donahue and Glasser data) – available?DWI data for the 5 chimps (Mars) – available?Macaque ex-vivo structural MRI data (Mars) – available?1 ex-vivo macaque (de Crespigny) – available?T1w/T2w myelin maps macaque (Mars?) – available?

We agree with the reviewer that sharing data and code are vitally important to increase the impact and benefit of our research and to increase reproducibility of our findings. In fact, much of the raw data that we used is already openly available via the National Chimpanzee Resource or the PRIME-DE database. All scripts that contain the relevant processing code will be uploaded in an openly accessible repository (git.fmrib.ox.ac.uk/neichert/project_MSM). This repository will also contain result workbench scene files, which allow interactive inspection of the data and group-level results.

Changes to manuscript text: As suggested by the reviewer, in order to make the sharing status of the data sets more easily identifiable for the reader, we updated our Data Availability Statement and added a Data Availability Overview table to the manuscript.